# Subfunctionalization of NRC3 altered the genetic structure of the *Nicotiana* NRC network

**Ching-Yi Huang**[1☯], **Yu-Seng Huang**[1☯], **Yu Sugihara**[2,3], **Hung-Yu Wang**[1], **Lo-Ting Huang**[1], **Juan Carlos Lopez-Agudelo**[1], **Yi-Feng Chen**[1], **Kuan-Yu Lin**[1], **Bing-Jen Chiang**[1], **AmirAli Toghani**[2], **Jiorgos Kourelis**[2,4], **Chun-Hsiung Wang**[5], **Lida Derevnina**[6], **Chih-Hang Wu**[1]*

1 Institute of Plant and Microbial Biology, Academia Sinica, Taipei, Taiwan, 2 The Sainsbury Laboratory, University of East Anglia, Norwich Research Park, Norwich, United Kingdom, 3 Iwate Biotechnology Research Center, Iwate, Japan, 4 Department of Life Sciences, Imperial College, London, United Kingdom, 5 Institute of Biological Chemistry, Academia Sinica, Taipei, Taiwan, 6 Crop Science Center, Department of Plant Sciences, University of Cambridge, Cambridge, United Kingdom

☯ These authors contributed equally to this work.

* wuchh@gate.sinica.edu.tw

**Data Availability Statement:** All data are available in the main text or the Supporting Information. Datasets associated with ancestral sequence reconstruction are available on Zenodo: 10.5281/zenodo.10354350 and 10.5281/zenodo.10360199.

## Abstract

Nucleotide-binding domain and leucine-rich repeat (NLR) proteins play crucial roles in immunity against pathogens in both animals and plants. In solanaceous plants, activation of several sensor NLRs triggers their helper NLRs, known as NLR-required for cell death (NRC), to form resistosome complexes to initiate immune responses. While the sensor NLRs and downstream NRC helpers display diverse genetic compatibility, molecular evolutionary events leading to the complex network architecture remained elusive. Here, we showed that solanaceous NRC3 variants underwent subfunctionalization after the divergence of *Solanum* and *Nicotiana*, altering the genetic architecture of the NRC network in *Nicotiana*. Natural solanaceous NRC3 variants form three allelic groups displaying distinct compatibilities with the sensor NLR Rpi-blb2. Ancestral sequence reconstruction and analyses of natural and chimeric variants identified six key amino acids involved in sensor-helper compatibility. These residues are positioned on multiple surfaces of the resting NRC3 homodimer, collectively contributing to their compatibility with Rpi-blb2. Upon activation, Rpi-blb2-compatible NRC3 variants form membrane-associated punctate and high molecular weight complexes, and confer resistance to the late blight pathogen *Phytophthora infestans*. Our findings revealed how mutations in NRC alleles lead to subfunctionalization, altering sensor-helper compatibility and contributing to the increased complexity of the NRC network.

## Author summary

Plants utilize complex immune systems to fend off invading pathogens. The nucleotide-binding domain and leucine-rich repeat (NLR) proteins function as intracellular immune receptors and play major roles in plant immunity. In solanaceous plants, several sensor

**Funding:** The work was supported by National Science and Technology Council (NSTC-110-2311-B-001-044, NSTC-111-2628-B-001-023, NSTC-112-2628-B-001-007 to CHW), intramural fund of Institute of Plant and Microbial Biology, Academia Sinica (CHW), National Institute of Agricultural Botany Fellowship (LD), the Gatsby Charitable Foundation (LD, YS, AT, JK), the Royal Society (LD), and BASF Plant Science (JK). The funders had no role in study design, data collection and analysis, decision to publish, or preparation of the manuscript.

**Competing interests:** I have read the journal's policy and the authors of this manuscript have the following competing interests: JK received funding from industry on NLR biology at the time of the study. JK has filed patents on NLR biology. Other authors have declared that no competing interests exist.

NLRs form a complex genetic network with helper NLRs called NRCs (NLR-required for cell death) to trigger immune responses. However, the evolution of genetic compatibility between sensor NLRs and NRCs was unclear. Here, we showed that NRC3, one of the NRC subgroups in solanaceous plants, underwent subfunctionalization after the *Solanum-Nicotiana* divergence, altering the NRC network in *Nicotiana*. Using natural, chimeric, and reconstructed ancestral NRC variants, we mapped six critical residues on multiple protein surfaces of NRC3 contributing to subfunctionalization. These findings reveal how mutations in NRC alleles lead to subfunctionalization, altering sensor-helper NLR compatibility and increasing the complexity of plant immune systems.

## Introduction

NLR (nucleotide-binding domain and leucine-rich repeat) proteins are intracellular receptors used by both plants and animals to detect invading pathogens [1,2]. They usually consist of an N-terminal domain that is essential for initiating downstream responses, an NB-ARC domain that binds ADP or ATP/dATP, and a leucine-rich repeat region that is involved in pathogen recognition [2]. Typical plant singleton NLRs, such as ZAR1, can recognize pathogen molecules and initiate downstream responses through the formation of a pentameric resistosome complex that functions as a calcium channel [3–5]. However, some NLR proteins have evolved into sensor and helper NLRs that function together, forming pairs or complex networks to confer resistance against invading microbes [2,6–8].

In asterids, the NRC (NLR-required for cell death) family represents a group of helper NLRs functioning downstream of multiple sensor NLRs [9–11]. In solanaceous plants, three of the NRCs, namely NRC2, NRC3, and NRC4, display partial genetic redundancy as well as specificity to different sensor NLRs, resulting in an NLR network with intricate genetic structure. For example, the sensor NLR Rpi-blb2 signals through NRC4 but not NRC2 and NRC3, the sensor NLR Prf signals through NRC2 and NRC3, whereas the sensor NLR Rx can signal redundantly through NRC2, NRC3 or NRC4 in the model solanaceous plant species *Nicotiana benthamiana* [9].

The NRC networks likely originated from a sensor-helper NLR gene cluster that emerged predating the divergence of asterids and Caryophyllales, and then massively expanded in lamiids, in particular in solanaceous plants and several *Ipomoea* species [9–11]. Among all the NRC family members described thus far, NRC0 is the only conserved NRC across lineages of asterid plants [10]. NRC0 orthologs from different species are often located in a gene cluster together with the sensor NLRs that are NRC0-dependent [10,11]. Many of the NRC0 subclade members can function with NRC0-dependent sensor NLRs from plants of other lineages, indicating that NRC0 orthologs are largely conserved and have partially retained their compatibility with sensor NLRs across different species [10,11]. Interestingly, the NRC networks are highly expanded in most lamiids, with several family-specific NRC subclades showing features of diversifying selection [11]. Most of these family-specific NRC members in lamiids do not function with tested sensor NLRs from different plant families, suggesting that the expansion of NRC networks has led to specialized pairings, resulting in low sensor-helper compatibility with NLRs from distantly related species [11].

While NLRs often show high diversity across plant species, many well-studied examples are NLRs that have remained largely conserved throughout evolution. For example, ZAR1, which represents an ancient category of plant immune receptors, indirectly recognizes effectors by engaging with its RLCK (Receptor-Like Cytoplasmic Kinase) partners, forming pentameric

resistosome complexes associated with the membrane [4,12]. Similar to ZAR1, helper NLRs, including ADR1, NRG1, and NRCs, form membrane-associated punctate and high molecular weight complexes upon activation by their respective sensor NLRs [13–18]. Recent research has uncovered that the activations of NRCs engage the conformational changes of resting homodimer complexes into hexameric resistosome complexes, providing valuable insights into the regulation of immunity conferred by the NRC network [19–22]. However, the exact mechanism by which the induction of the NRC resistosome complex engages with the coordination of both sensor and helper NLRs, as well as how the compatibility between these sensor-helper NLRs is determined in the network, remains unclear. As the sensor and helper NLRs within the NRC network trace back to a common ancestral sensor-helper cluster similar to other NLR pairs, it is intriguing to observe that diverse levels of compatibility have evolved among multiple helper NLRs and sensor NLRs [10,11].

In this study, we addressed a fundamental feature of the NRC network architecture: what are the molecular determinants of sensor/helper specificity? We focused on the evolution of specificity between NRC3 orthologs to the sensor NLR Rpi-blb2. We found that Rpi-blb2 can only signal (referred to as compatible from here on) through a subset of NRC3 variants. Using ancestral sequence reconstruction, we showed that the change of compatibility evolved through subfunctionalization. We mapped the determinants that affect the compatibility of NRC3 orthologs with Rpi-blb2 to three residues in the NB-ARC domain and three residues in the LRR domain. These residues are positioned at three surfaces on the resting NRC3 homodimer, and collectively contribute to their compatibility with Rpi-blb2. We propose that the NRC network evolved through successive cycles of helper NLR duplications followed by mutations leading to subfunctionalization. These subfunctionalization events may divide the NLR network into smaller subnetworks, increasing the complexity of the immune system.

## Results

### NRC3 orthologs and paralogs show different compatibility with the sensor NLR Rpi-blb2

To gain insights into the molecular mechanisms and evolution of the NRC network, we cloned several NRC homologs from tomato and *N. benthamiana* and performed complementation assays with several sensor NLRs in the *nrc2/3/4* CRISPR knockout (*nrc2/3/4*_KO) *N. benthamiana* line [23]. We reasoned that exploring closely related NRC homologs that show different compatibilities to the sensor NLRs may shed light on the molecular determination and evolution of the sensor-helper compatibility. The cloned NRCs were grouped into NRC0 to NRCX based on the phylogenetic analysis (S1 Fig) [9,24–26]. We individually expressed these NRCs with Rx, Sw5b, Prf (Pto/AvrPto), Gpa2, Rpi-blb2, and R1 with their corresponding effector proteins in *N. benthamiana* leaves, and performed cell death intensity quantification using autofluorescence-based imaging (S2A Fig). While most of the sensor-helper genetic dependency results were consistent with the previous report, tomato NRC3 (SlNRC3) but not *N. benthamiana* NRC3 (NbNRC3) rescued Rpi-blb2-mediated cell death in the *nrc2/3/4*_KO *N. benthamiana* (Figs 1A, 1B, and S2B–S2F).

To explore the differences among the NRC3 variants, we performed phylogenetic analyses of several NRC3 sequences identified from solanaceous plants. We found that most solanaceous NRC3 homologs are clustered into three allelic groups, including Group A (NRC3a) which contains orthologs of NRC3 from *Solanum* and *Capsicum* species, and Group B (NRC3b) and Group C (NRC3c) which contains NRC3 from *Nicotiana* species (Fig 1C). Both *N. tabacum* and *N. sylvestris* harbor sequences of NRC3b and NRC3c, whereas *N. benthamiana* only contains one NRC3c sequence (Fig 1C). We cloned several of these NRC3 variants

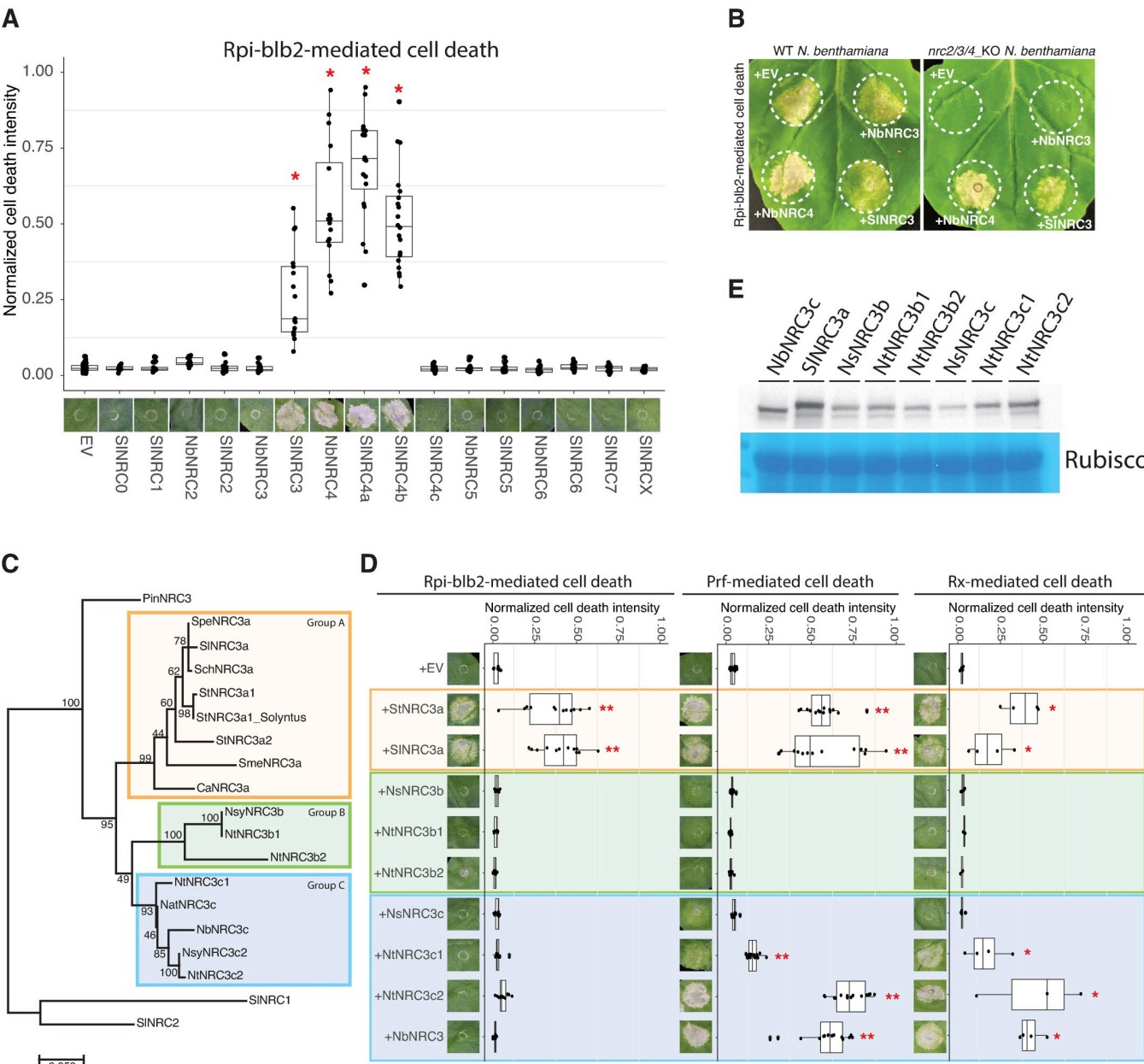

**Fig 1. Rpi-blb2 signals through NRC3a but not NRC3b or NRC3c. (A)** Cell death assays of Rpi-blb2 with different NRCs. Rpi-blb2 and AVRblb2 were co-expressed with indicated NRCs cloned from tomato and *N. benthamiana* in *nrc2/3/4_KO N. benthamiana*. Cell death intensity and phenotypes were recorded at 6 dpi. The line in the boxplots represents the medium, the box edges represent the 25th and 75th percentiles, and the whiskers extend to the most extreme data points no more than 1.5x of the interquartile range. Statistical differences between the negative control (EV) and tested groups were examined by paired Wilcoxon signed rank test (* = p < 0.0001). **(B)** Assays of Rpi-blb2-mediated cell death rescued by NRC variants. Rpi-blb2 and AVRblb2 were co-expressed with NRCs as indicated in both WT or *nrc2/3/4_KO N. benthamiana*. **(C)** Phylogenetic analysis of NRC3 natural variants identified from tomato, tobacco potato, pepper, and eggplant. Sequence alignment of the NB-ARC domain was used to generate the phylogenetic tree using the Maximum likelihood method with 1000 bootstrap tests. SlNRC1 and SlNRC2 were selected as outgroups. The scale bars indicate the evolutionary distance in amino acid substitution per site. The orange, green, and blue boxes indicate allelic groups A, B, and C, respectively. **(D)** Cell death assays of different sensor NLRs with NRC3 natural variants. The cloned NRC3 natural variants were co-expressed with Rpi-blb2/AVRblb2, Pto/AvrPto, or Rx/CP in *nrc2/3/4_KO N. benthamiana*. **(E)** Protein accumulation of NRC3 natural variants. NRC3 natural variants were transiently expressed in WT *N. benthamiana*. The proteins were extracted from leaf samples at 2 dpi and the NRC3 protein accumulations were detected by α-myc antibody. SimplyBlue SafeStain-staining of Rubisco was used as the loading control. The dot plots represent cell death intensity quantified by UVP ChemStudio PLUS at 6 dpi. The line in the boxplots represents the medium, the box edges represent the 25th and 75th percentiles, and the whiskers extend to the most extreme data points no more than 1.5x of the interquartile range. Statistical differences between the negative control (EV) and tested groups were examined by paired Wilcoxon signed rank test (* = p<0.05, ** = p < 0.0001).

and tested their ability to rescue Rpi-blb2, Prf (Pto), and Rx-mediated cell death in the *nrc2/3/4*_KO *N. benthamiana*. We found that the two NRC3a variants tested were able to rescue Rpi-blb2, Prf, and Rx-mediated cell death, while the NRC3b variants failed to rescue cell death mediated by these sensors. Interestingly, most NRC3c variants were able to function with Prf and Rx but not Rpi-blb2 (Fig 1D). The accumulations of all the natural variants were detectable, and none of them showed strong auto-activity in inducing cell death when expressed alone (Figs 1E and S3). These results suggested that NRC3 homologs have evolved to be functionally divergent.

## Ancestral reconstructions reveal subfunctionalization of NRC3c towards loss of compatibility with Rpi-blb2

To determine which molecular events contributed to the functional divergence of NRC3, we performed functional assays of reconstructed ancestral NRC3 variants. We extracted 324 non-redundant nucleotide sequences of the *NRCX*, *NRC1*, *NRC2*, and *NRC3* clades from 124 solanaceous genomes, and then used FastML to reconstruct the ancestral NRC sequences (S4A Fig). We synthesized the full-length of five ancestral *NRC3* variants reconstructed from the FastML, including *N4*, *N95*, *N89*, *N88*, and *N3* that represent the ancestral state of *NRC3a*, *NRC3b*, *NRC3c*, *NRC3b/c*, and before the divergence of the three allelic groups (Figs 2A and S5). We then tested the degree to which these ancestral variants can rescue Rpi-blb2, Prf (Pto), and Rx-mediated cell death. We found that while N3, N4, and N88 rescue Rpi-blb2-mediated cell death, the ancestral variants N95 (NRC3b) and N89 (NRC3c) show no or low activities in rescuing Rpi-blb2-mediated cell death (Fig 2B). While most of these ancestral variants (N3, N4, N88, and N89) rescued Prf (Pto) and Rx-mediated cell death, N95 was the only variant that failed to rescue any cell death tested (Fig 2B). The accumulations of all of these ancestral variants were detectable with low or no auto-activity (S6 Fig). We introduced a D to V mutation into the MHD motif of N88, N95 and NRC3b variants, and found that, while N88$^{DV}$ induced very strong cell death, none of the N95$^{DV}$ and NRC3b$^{DV}$ variants induced cell death in *N. benthamiana*, suggesting that this group of NRC3 has nonfunctionalized during the evolutionary process (Fig 2C). These results indicate that NRC3 in the ancestral species likely functions together with Rpi-blb2/Prf/Rx, whereas the NRC3 variants that evolved in *Nicotiana* species acquired mutations leading to nonfunctionalization (NRC3b) or subfunctionalization (NRC3c), losing their ability to work together with Rpi-blb2.

To further test this hypothesis, we synthesized *NRC3* of *Petunia inflata* (*PinNRC3*) which is sister to the three *NRC3* allelic groups mentioned above (Fig 1C). We noticed that the PinNRC3 from the genome database contains an indel of 9 amino acids between the CC and NB-ARC domains (S7A Fig). Therefore, we manually curated the sequence by inserting 9 amino acids from SlNRC3a into the indel of PinNRC3, and found that this manually curated PinNRC3 variant is able to rescue all three cell death phenotypes tested (S7B Fig). Taken together, these results support the hypothesis that the ancestral NRC3 can function with a broader collection of sensor NLRs, while NRC3c variants subfunctionalized to work with a smaller subset of sensor NLRs.

## Six NRC3 amino acid residues contribute to the compatibility of NRC3 with Rpi-blb2

Since SlNRC3a and NbNRC3c showed robust differences in their ability to function with Rpi-blb2, we focused on these two NRC3 variants to identify the residues that affect their compatibility with Rpi-blb2. Pairwise sequence comparison indicated that the two NRC variants share around 86.42% overall sequence identity, with 87%, 90%, and 83% for the CC domain,

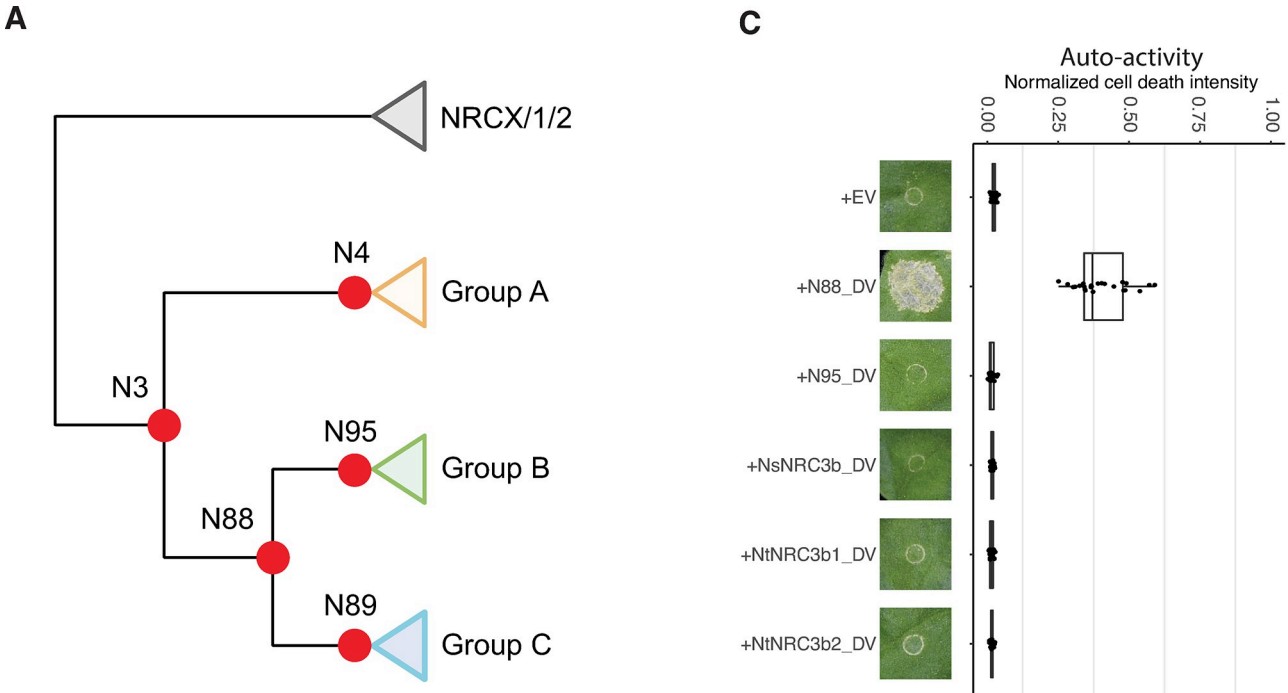

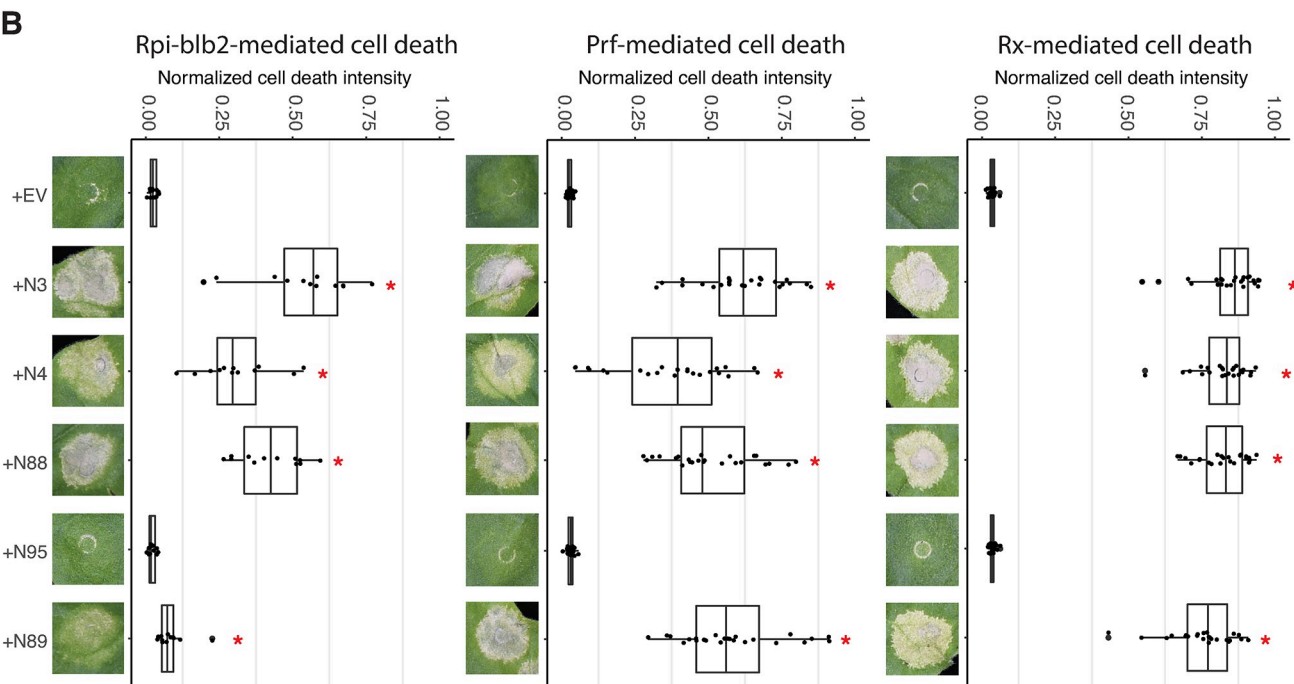

**Fig 2. Subfunctionalization contributes to the evolution of NRC3c.** (**A**) Phylogenetic tree of NRC3 natural variants. Orange, green, and blue boxes represent allelic groups A, B, and C, respectively. Red dots indicate the nodes of reconstructed ancestral NRC3 variants. (**B**) Cell death assays of ancestral NRC3 variants. The ancestral variants were co-expressed with Rpi-blb2/AVRblb2, Pto/AvrPto, or Rx/CP in *nrc2/3/4*_KO *N. benthamiana*. (**C**) Cell death analysis of NRC3b_DV, N88_DV and N95_DV. The NRC3_DV variants carry a D to V mutation in the MHD motif. These variants were expressed alone in WT *N. benthamiana*. The dot plots represent cell death intensity quantified by UVP ChemStudio PLUS at 6 dpi. The line in the boxplots represents the medium, the box edges represent the 25th and 75th percentiles, and the whiskers extend to the most extreme data points no more than 1.5x of the interquartile range. Statistical differences between the negative control (EV) and tested groups were examined by paired Wilcoxon signed rank test (* = p < 0.0001).

NB-ARC domain, and LRR domain, respectively (Fig 3A). We then generated chimeric proteins between the two NRC3 variants and determined which regions contribute to the compatibility with Rpi-blb2. These constructs were named NNS, NSN, NSS, SSN, SNS, and SNN (N stands for *N. benthamiana*; S stands for *S. lycopersicum*) (S8A Fig). The chimeric variants carrying the CC domain from SlNRC3a (SNS, SSN, SNN) showed no or low activity in rescuing Rpi-blb2-mediated cell death, suggesting that the CC domain is not critical in determining their compatibility with Rpi-blb2 (S8A Fig). While the chimeric variants NSN and NNS showed low activities, the variant NSS fully rescued Rpi-blb2-mediated cell death, similar to the degree observed using SSS (SlNRC3) (S8A Fig). None of these chimeric NRC3 variants were auto-active, with SNS being the only variant that failed to rescue Prf-mediated cell death (S8A Fig). All of these NRC3 chimeric variants were detectable using immunoblot analysis (S8B Fig). These results suggest that the NB-ARC and LRR domains cooperatively determine the compatibility of NRC3 variants with Rpi-blb2.

To further identify the key polymorphisms within the NB-ARC and LRR domains that contribute to sensor-helper compatibility, we generated two sets of chimeric variants of Nb/SlNRC3 for complementation assays, focusing on the polymorphisms in either the NB-ARC or the LRR domain. We divided the NB-ARC domain into 10 smaller regions based on sequence alignment and generated corresponding chimeric NRC3 variants named $NN_{1-10}S$ (Figs 3B and S9). While most of the $NN_{1-10}S$ chimeric NRC3 variants showed low activity in rescuing Prf-mediated cell death, the variant $NN_3S$ effectively rescued Prf-mediated cell death and fully rescued Rpi-blb2-mediated cell death (Figs 3B and S9). This region contained six amino acid differences between the two variants (Fig 3B). To identify the regions in the LRR domain that contribute to sensor-helper compatibility, we generated chimeric protein $NSN_{1-5}$, with polymorphisms from the LRR of tomato NRC3 swapped into the NSN background (Fig 3C). While the NRC3 variant $NSN_1$ was auto-active (S10A Fig), the variant $NSN_2$ and $NSN_5$ both partially rescued Rpi-blb2-mediated cell death in the *nrc2/3/4_KO N. benthamiana* (Figs 3C and S10). Thus, we generated another chimeric NRC3 variant, named $NSN_{25}$, and found that this variant fully rescued Rpi-blb2 cell death (Fig 3C). These results suggest that both the NB-ARC and LRR domains cooperatively enable the NRC3 variant to function with Rpi-blb2.

To pinpoint the key residues in region 2 of the LRR domain, we tested additional chimeric NRC3 variants. We found that a single amino acid change (I to T) at position 642 is sufficient to enable NSN to weakly rescue Rpi-blb2-mediated cell death (S11A and S11C Fig). This single amino acid change also conferred full activity in rescuing Rpi-blb2 cell death in the $NSN_5$ background (Figs 3D, S11B and S11D). We divided region 5 of the LRR domain into five fragments and generated variants named $NSN_{5a-e}$, and found that $NSN_{5b}$ and $NSN_{5e}$ partially rescue Rpi-blb2-mediated cell death (S12A and S12C Fig). We combined these polymorphisms into the $NSN_2$ background to generate a new variant named $NSN_{25be}$. As expected, the NRC3 variant $NSN_{25be}$ fully rescued Rpi-blb2-mediated cell death (Figs 3E, S12B and S12D).

We sought to pinpoint the amino acid residues in each region that enabled NRC3 variants to function with Rpi-blb2 in the NbNRC3c background. We introduced the polymorphisms identified above into NbNRC3c, generating a chimeric variant named $NN_3N_{T5be}$. We found that $NN_3N_{T5be}$ can fully rescue cell death mediated by Rpi-blb2 in *nrc2/3/4_KO N. benthamiana* (Figs 3F and S13). Through additional sets of loss-of-function screens, we found that amino acid changes of S202P, T203K, N221K, C824H, N832K, and V881I quantitatively affect the ability of NRC3 to function with Rpi-blb2 (Figs 3G and S14). Thus, we introduced these mutations into NbNRC3c together with I642T, and found that this new NRC3 variant, named $NN_{PKK}N_{THKI}$, can fully rescue Rpi-blb2-mediated cell death (Figs 3H and S15). Since V and I at position 881 are similar residues, we further generated $NN_{PKK}N_{THK}$ and found that this variant showed similar activity to $NN_{PKK}N_{THKI}$ (Fig 3H). Based on the above results, we

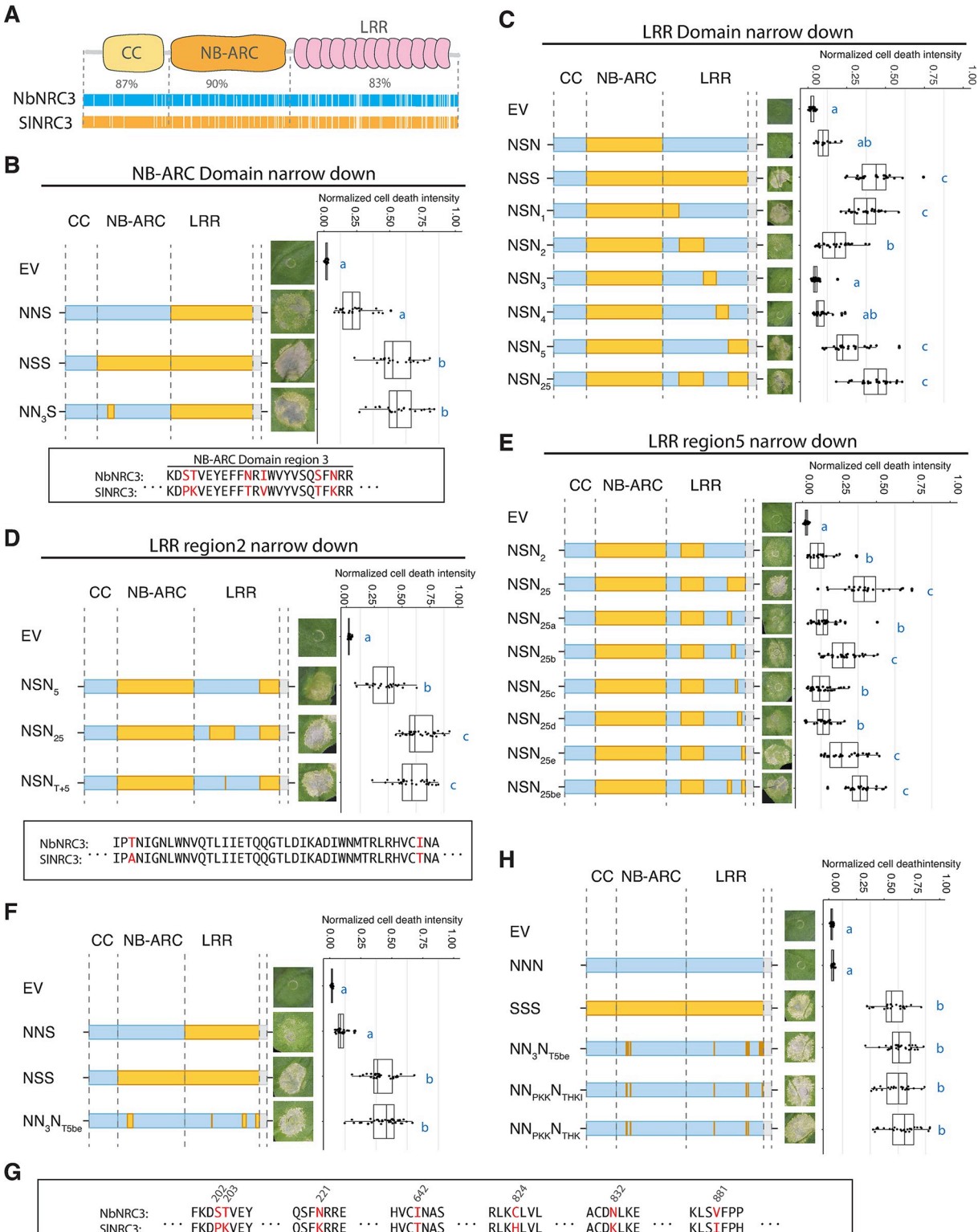

**Fig 3. Changing six residues in NbNRC3 enabled it to function with Rpi-blb2. (A)** Sequence identity between NbNRC3 (NNN) and SlNRC3 (SSS) in CC, NB-ARC, and LRR domains. Cell death assays of chimeric NRC3 variants designed to investigate polymorphisms in **(B)** the NB-ARC domain, **(C)** the LRR domain, **(D)** LRR domain region 2, and **(E)** LRR domain region 5 that contribute to the sensor-helper compatibility. **(F)** Cell death assays of chimeric NRC3 variants that carry polymorphisms identified in (B-E). **(G)** Polymorphisms of NbNRC3 and SlNRC3 at the seven positions identified (highlighted in red). **(H)** Cell death assays of chimeric NRC3 variants NN$_{PKK}$N$_{THKI}$ and

$NN_{PKK}N_{THK}$. Cell death assays were performed by co-expressing chimeric NRC3 variants with Rpi-blb2/AVRblb2 in *nrc2/3/4*_KO *N. benthamiana*. In the schematic representations of the NRC3 variants, the blue color indicates the regions/residues from NbNRC3 (N), and the orange color indicates the regions/residues from SlNRC3 (S). The dot plots represent cell death intensity quantified by UVP ChemStudio PLUS at 6 dpi. The line in the boxplots represents the medium, the box edges represent the 25th and 75th percentiles, and the whiskers extend to the most extreme data points no more than 1.5x of the interquartile range. Statistical differences were examined using Dunn's test ($p < 0.05$).

concluded that changes of 6 amino acids, three in the NB-ARC domain and three in the LRR domain, enable NbNRC3c to function with Rpi-blb2. These results suggest that changes in amino acids at these positions pose a major contribution to sensor-helper compatibility and subfunctionalization during the evolution of the NRC3c allelic group.

## Two K to N mutations in the NB-ARC and LRR domains play critical roles in NRC3 subfunctionalization

Next, we looked into the polymorphism of these 6 positions in the NRC3 natural variants mentioned above (Figs 1C and 3G). We found that all the NRC3 variants from allelic group A possess PKKTHK, and all the variants from allelic group B and the outgroup PinNRC3 possess PKKTRK (Fig 4A). Interestingly, sequences from allelic group C showed higher diversity, with NbNRC3c being the most diverse variant (Fig 4A). We introduced these different polymorphisms into the NbNRC3c background and tested the ability of these variants to rescue Prf and Rpi-blb2 cell death in *nrc2/3/4*_KO *N. benthamiana*. While all these variants rescued Prf-mediated cell death, NRC3 variants with PKKTHK or PKKTRK, but not PKNTRN or PTNTRN, were able to fully rescue Rpi-blb2-mediated cell death (Figs 4A and S16). Consistent with the results from the polymorphisms found in the natural variants, the major differences between N88 (PKKTRK) and N89 (PKNTRN) were also the two K to N mutations (Figs 3G and 4B). These findings indicate that the two mutations converting K to N are likely the most crucial changes during the subfunctionalization process. To further test this hypothesis, we introduce the two K to N mutations into the ancestral variants N88 ($N88^{NN}$) or the N to K mutations into the ancestral variants N89 ($N89^{KK}$). We found that $N88^{NN}$ showed reduced activity and $N89^{KK}$ showed increased activity compared to their ancestral states respectively (Figs 4B and S17). These results support the finding that six amino acid residues contribute to the compatibility of NRC3 variants to Rpi-blb2, with two K to N changes playing major roles in the NRC3 subfunctionalization process.

To understand the polymorphisms of these two positions across NRC3 alleles, we performed entropy analysis using the protein sequence alignment of the natural NRC3 variants. We found that these positions both had entropy values of 0.637 and did not stand out as being among the most conserved or diversified positions (Fig 4C and S4 Dataset). We then calculated the dN-dS value using SLAC (Single-Likelihood Ancestor Counting) and found that the residue at position 221 showed a slightly higher dN than dS value, while the residue at position 832 showed a lower dN value than dS value (Fig 4D and S5 Dataset). Despite this, the FEL (Fixed Effects Likelihood) analysis indicated that both positions are under neutral selection (Fig 4D and S5 Dataset). These results suggest that variations in these amino acids among different NRC3 allelic groups, leading to subfunctionalization, are more likely the outcome of random genetic drift rather than the consequence of strong selection.

## Sensor-helper compatibility is determined by multiple protein surfaces

To understand the spatial arrangement of the residues involved in sensor-helper compatibility on the NRC3 structure, we performed predictions using AlphaFold2 and fitted the structure model onto the recently published NRC2 resting homodimer (S18 Fig) [19]. We then

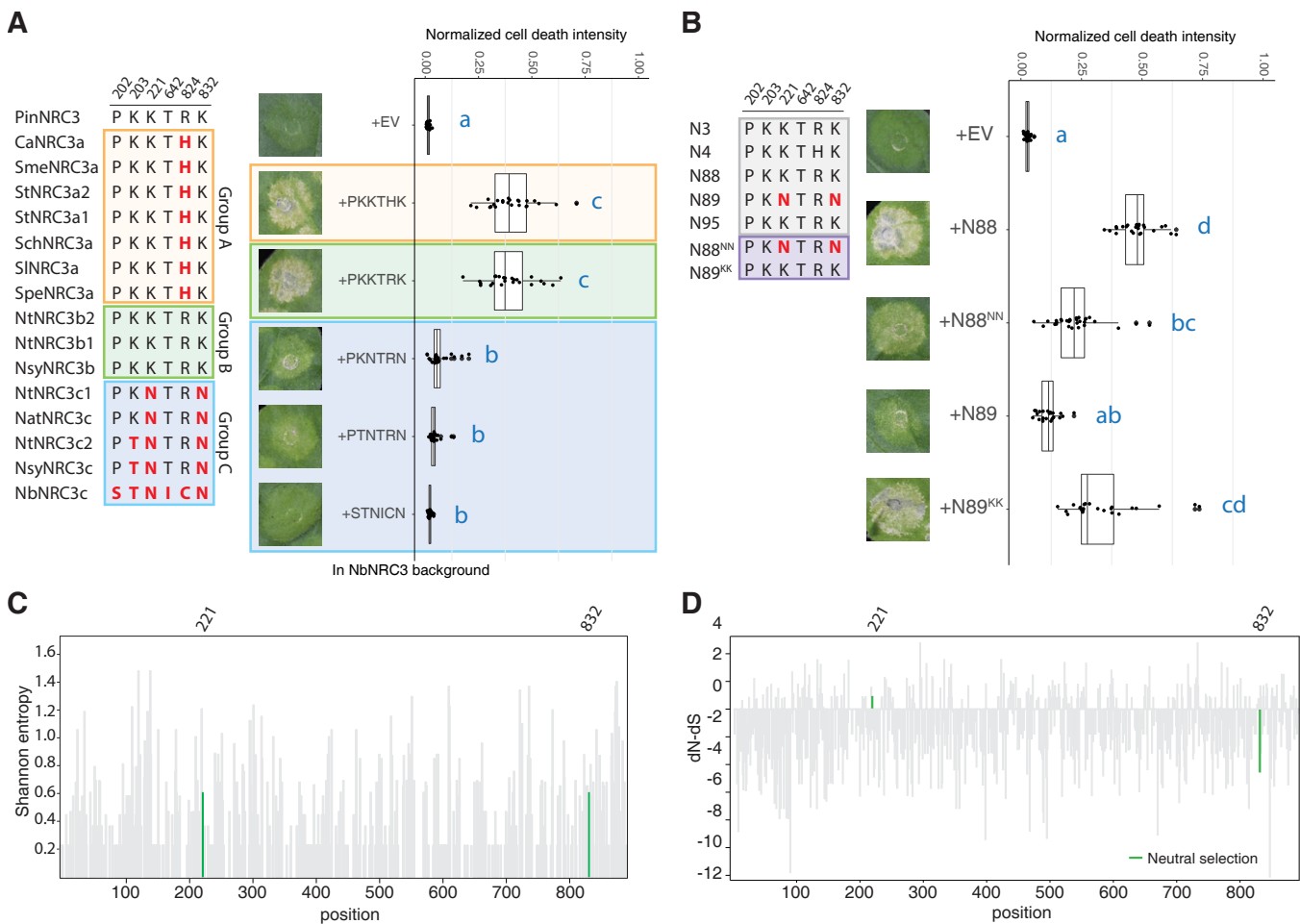

**Fig 4. Two K to N mutations in the NB-ARC and LRR domains play major roles in NRC3 subfuntionalization.** (A) Left panel, the polymorphisms in the NRC3 natural variants at the 6 positions identified. Right panel, cell death assay testing these polymorphisms in NbNRC3c background. Introducing PKKTHK or PKKTRK enabled it to function with Rpi-blb2. (B) Left panel, the polymorphisms in the ancestral NRC3 variants at the 6 positions identified. Right panel, cell death assay testing two lysine-asparagine changes in both N88 and N89 backgrounds. Swapping two K to N in N88/N89 changed the compatibility to Rpi-blb2. The dot plots in (A) and (B) represent cell death intensity quantified by UVP ChemStudio PLUS at 6 dpi. The line in the boxplots represents the medium, the box edges represent the 25th and 75th percentiles, and the whiskers extend to the most extreme data points no more than 1.5x of the interquartile range. Statistical differences were examined using Dunn's test (p < 0.05) (C) The entropy analysis of natural NRC3 variants. The protein sequences of NRC3s were aligned using MAFFT and the Shannon entropy was calculated. The positions of the two K to N changes were highlighted in green. (D) dN-dS calculation of natural NRC3 variants by using SLAC analysis. The positions of K to N changes were highlighted in green. The two residues were under neutral selection based on FEL analysis.

highlighted the residues involved in sensor-helper compatibility on the predicted resting NbNRC3c homodimer complex (Fig 5A). We found that N832(K), located near the end of the LRR, is in proximity to S202(P) and T203(K) in the NB-ARC domain, suggesting they may be on the same exposed surface of the resting complex of NRC3 (Fig 5B). Both of the residues I642(T) and C824(H) are on the concave surface of the LRR domain, facing a cavity in between the LRR and the NB-ARC domain (Fig 5C). Interestingly, the residue N221(K) is located in the region between the two NbNRC3c protomers, corresponding to interface 1a described in the NRC2 homodimer (Fig 5D) [19]. This residue is also positioned next to the cavity in between the LRR and the NB-ARC domain (Fig 5C).

To further investigate whether the three surfaces collectively contribute to sensor-helper compatibility, we generated variants containing residues from SlNRC3 in only two of the

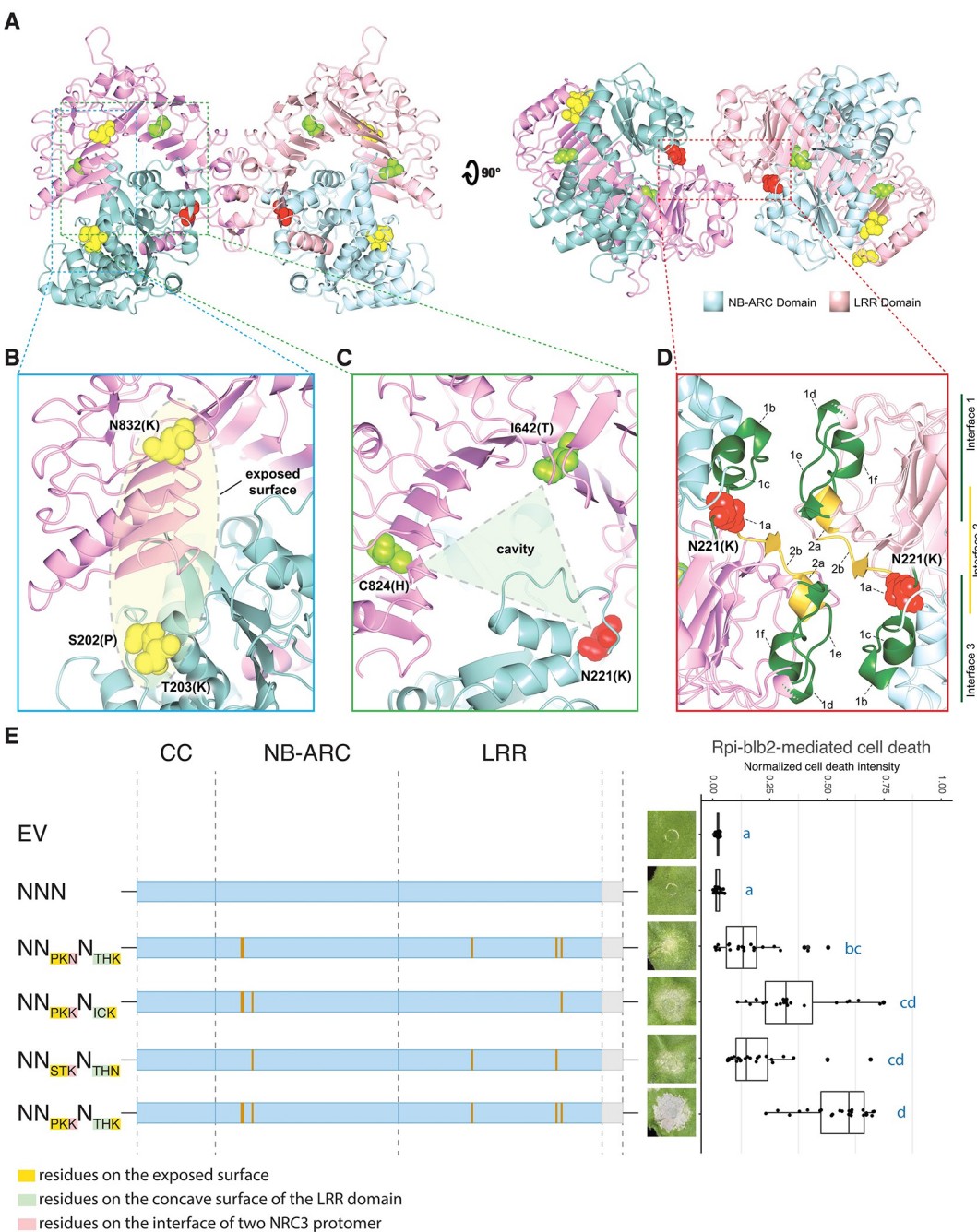

**Fig 5. Multiple protein surface contribute to the sensor-helper compatibility determination.** (A) Structure of NbNRCc homodimer shown in two orthogonal views. The NB-ARC domains of two protomers are shown in light blue and the LRR domains are shown in pink. The 6 residues involved in sensor-helper compatibility are highlighted in yellow, green, and red. (B) Details of the exposed surface between S202(P)/T203(K) and N832(K). (C) Details of the cavity in between the NB-ARC domain and the LRR domain. This cavity is surrounded by I642(T), C824(H), and N221(K). (D) Details of the interface between two NbNRC3c protomers where the N221(K) is located. The interfaces are highlighted based on the resting NbNRC2 homodimer. (E) Cell death assays of NRC3 variants carrying identified residues from SlNRC3 in only two of the surfaces. The residues on the exposed surface, on the concave surface of the LRR domain, and on the interface between NRC3 protomers are highlighted in yellow, green, and red, respectively. Cell death assays were performed by co-expressing chimeric NRC3 variants with Rpi-blb2/AVRblb2 in *nrc2/3/4*_KO *N. benthamiana*. The dot plots represent cell death intensity quantified by UVP ChemStudio PLUS at 6 dpi. The line in the boxplots represents the medium, the box edges represent the 25th and 75th percentiles, and the whiskers extend to the most extreme data points no more than 1.5x of the interquartile range. Statistical differences were examined using Dunn's test (p < 0.05).

surfaces using NNN as the background (Fig 5E). Compared to the $NN_{PKK}N_{THK}$ variant, all these three new variants ($NN_{PKN}N_{THK}$, $NN_{PKK}N_{ICK}$, and $NN_{STK}N_{THN}$) only showed partial activities in rescuing Rpi-blb2-mediated cell death (Fig 5E). All of these variants rescued Prf-mediated cell death, with no auto-activity, and accumulated to a similar level (S19 Fig). These results suggested that the three surfaces on NRC3 contribute to the sensor-helper compatibility collectively.

## Steady-state interactions between sensor and helper NLRs do not reflect their compatibility

To further dissect the molecular mechanisms of sensor-helper compatibility, we tested the interactions between NRC3 variants with the sensor NLR Rpi-blb2. We focused on two variants, NNN (NbNRC3c) and $NN_{PKK}N_{THK}$, as these two variants differ from each other by only six amino acids but display robust differences in their compatibility with Rpi-blb2 (Fig 3H). We co-expressed Rpi-blb2 with NNN or $NN_{PKK}N_{THK}$ with or without AVRblb2 and then performed co-immunoprecipitation analyses. When we pulled down Rpi-blb2, we detected very weak signals from NNN and $NN_{PKK}N_{THK}$ regardless of whether AVRblb2 was present or not (S20A Fig). Similarly, when we pulled down the two NRC3 variants, we detected weak signals from Rpi-blb2 (S20B Fig). These results suggest that steady-state interactions between sensor and helper NLRs of the NRC superclade detected using co-IP experiments do not reflect their compatibility.

## Introducing the 6 mutations into NbNRC3c enabled it to form resistosomes upon activation and confer resistance against *Phytophthora infestans*

Recent studies suggest that NRCs form membrane-associated punctate upon activation by corresponding sensor NLRs and effectors [15,27]. To test whether the compatibility of NRC3 variants to the sensor NLRs is consistent with punctate formation, we performed cell biology assays by co-expression of NRC3 variants with Rpi-blb2/AVRblb2 or Pto/AvrPto. Co-expressions with Pto/AvrPto triggered both NbNRC3c and SlNRC3a to form membrane-associated punctate, whereas Rpi-blb2/AVRblb2 only triggered SlNRC3a but not NbNRC3c to form membrane-associated punctate (Figs 6A, 6B and S21). Consistent with these findings, Rpi-blb2/AVRblb2 can also trigger NRC3 variants NSS, $NN_3N_{T5be}$, and $NN_{PKK}N_{THK}$ to form membrane-associated punctate (Figs 6A, 6B and S21). We used BN-PAGE (Blue Native Polyacrylamide Gel Electrophoresis) to analyze the complex size of NRC3 variants co-expressed with Rpi-blb2 in the presence or absence of AVRblb2. In the absence of AVRblb2, both NNN and $NN_{PKK}N_{THK}$ variants migrated as dimers. In the presence of AVRblb2, the variant NNN still migrated as dimer while the variant $NN_{PKK}N_{THK}$ migrated mostly as high molecular weight complexes with reduced dimer form (Fig 6C). Furthermore, NRC3 variants, including SlNRC3a and $NN_{PKK}N_{THKI}$, that functioned together with Rpi-blb2 in the cell death assays, also rescued Rpi-blb2-mediated disease resistance to late blight pathogen *Phytophthora infestans* in the *nrc2/3/4_KO N. benthamiana* (Fig 6D).

## Discussion

The NRC network is important for pathogen resistance in solanaceous crops, but the determinants and evolution of its complex genetic structure have been elusive. Here, by exploring ancestral and natural variants of NRC homologs, we revealed that subfunctionalization is key to the evolution of the NRC network. By examining closely related NRC3 orthologs, we

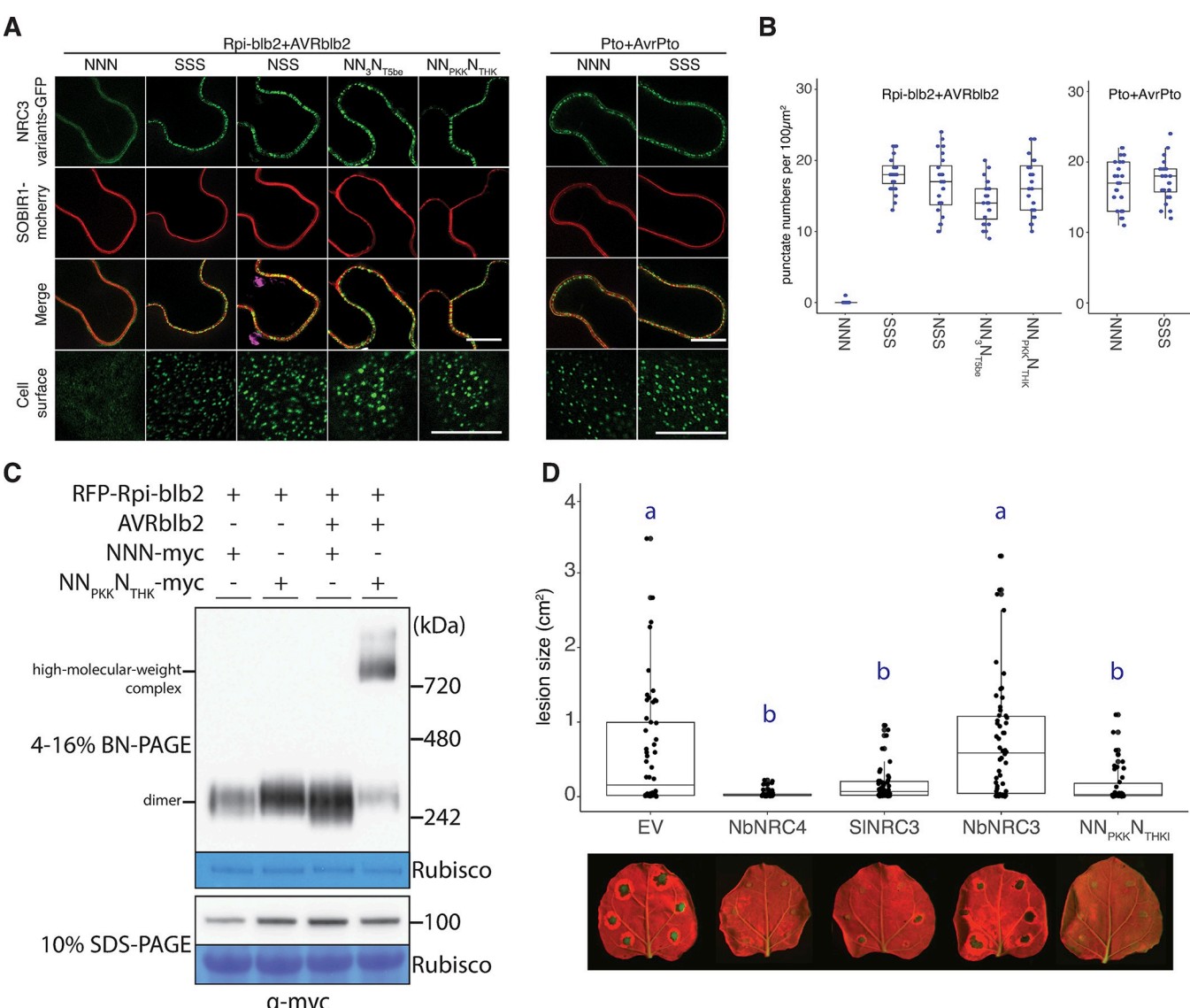

**Fig 6. Introducing the 6 mutations into NbNRC3c enabled it to form resistosomes upon activation, and to confer resistance against *P. infestans*.** (A) Compatible NRC3 variants form membrane-associated punctate upon immune activation. NRC3 variants fused with GFP were co-expressed with Rpi-blb2/ AVRblb2 or Pto/AvrPto. Samples were examined at 3 dpi. Scale bars represent 10μm. SOBIR1-mcherry was used as a plasma membrane marker. (B) Quantification of punctate of NRC3 variants in (A). (C) BN-PAGE analyzing the complex size of NRC3 variants. The NRC3 variants were co-expressed with Rpi-blb2 in the presence or absence of AVRblb2 in *nrc2/3/4*_KO *N. benthamiana*. Leaf samples were collected at 2 dpi. (D) Infection assay of *P. infestans* in *nrc2/3/4*_KO *N. benthamiana* transiently expressing Rpi-blb2 and NRC variants. The dot plots represent the lesion sizes observed at 5 days post-infection quantified by UVP ChemStudio PLUS. The line in the boxplots represents the medium, the box edges represent the 25th and 75th percentiles, and the whiskers extend to the most extreme data points no more than 1.5x of the interquartile range. Statistical differences were examined using Tukey's HSD test (p < 0.05).

identified critical polymorphisms in sensor-helper compatibility, pinpointing three amino acids in both the NB-ARC and LRR domains. Notably, two lysine-to-asparagine mutations significantly impact sensor-helper compatibility and subfunctionalization. Our findings thus unravel both mechanistic and evolutionary aspects underpinning the complex genetic architecture of the NRC network.

How did subfunctionalization events impact the evolution of NLR networks? We propose that the subfunctionalization of NRCs led to the division of the NLR network into smaller sub-networks (S22 Fig). However, evolutionary analyses indicated that residues that affect NRC3

compatibility to Rpi-blb2 are mostly under neutral selection, suggesting that subfunctionalization may be the result of random genetic drift accumulating over time rather than being due to strong selection. The evolution from ancestral NRC sensor-helper gene cluster to the complex network with partially redundant NRC2/3/4 nodes in solanaceous plants may have been the result of successive cycles of gene duplications followed by subfunctionalizations (S22 Fig). Interestingly, NRC networks are highly expanded independently in several lamiids, with many NRCs showing partial redundancy, such as in both Solanaceae and Convolvulaceae [10,11]. NRC networks with intricate genetic architectures must have provided some long-term advantages for the plant species to survive. One possibility is that this helped improve the robustness of the immune system, becoming more resistant to pathogen perturbation. Such examples include SPRYSEC15 and AVRcap1b effectors that target NRC2 and NRC3, but not NRC4 [28]. If little or no diversification and subfunctionalization occurs among NRC homologs, a single or few effectors that suppress the function of NRCs could have dramatically compromised the NRC-mediated immunity.

The difference in signaling compatibilities between orthologous helper and sensor NLRs has been observed in many other cases. Since the NRC network originated from an ancestral NLR pair, the phenomenon observed in paired NLRs may resemble the early scenario in NRC network evolution. In Arabidopsis, two orthologous RPS4/RRS1 NLR pairs were reported. Members in these two pairs, RPS4A/RRS1A and RPS4B/RRS1B, only function with their authentic partners [29]. While the TIR domain is indispensable for the specificity between RPS4A/RRS1A and RPS4B/RRS1B pairs, swapping the NB-ARC, LRR, or DOM4 of RRS1B into RRS1A all compromised the cell death responses when co-expressed with RPS4A and the effector [30]. In rice, the helper NLR Pias-1 can also function with the sensor NLR RGA5 to respond to AVR-Pia, indicating that Pias-1 is not specialized to its own linked sensor NLR Pias-2 [31]. In contrast, sensor-helper specialization was observed in allelic Pik-1/Pik-2 rice NLR pairs [32]. For Pikm-1/Pikm-2 and Pikp-1/Pikp-2, this sensor-helper specialization was mapped to a single amino acid polymorphism on Pik-2 that determines its preferential association between matching pairs [33]. Presuming that orthologous paired NLRs should exhibit cross-compatibility immediately following the initial duplication, various degrees of subfunctionalization have occurred throughout the co-evolution of distinct paired NLRs and their orthologous counterparts.

Our research primarily examined mutations in helper NLRs and their influence on compatibility with sensor NLRs. It is plausible that sensor NLRs also possess corresponding residues dictating their compatibility with helper NLRs. These residues might determine the ability of sensor NLRs to signal through a broader or narrower array of NRCs. Notably, the sensor NLR Rx is distinguished by its capacity to signal through a wider range of NRCs compared to other tested sensor NLRs, functioning with solanaceous NRC2/3/4 and multiple NRCs in *Ipomoea aquatica* [11]. The NB-ARC domain of Rx is sufficient to disrupt the resting NRC dimer and induce NRC oligomerization to trigger immune signaling, highlighting the key role of NB-ARC domain in sensor-helper compatibility from the sensor perspective [19,34]. Investigating additional natural variants and analyzing the residues involved in sensor-helper NLR compatibility across different sensor and helper NLRs could further elucidate how the signaling connections in the NRC network maintain their specificity and promiscuity.

Finally, another intriguing aspect yet to be fully explored is the signal transduction mechanism from sensor to helper NLRs. The activation-and-release model has been proposed to explain helper NLR activation by sensor NLRs following pathogen effector detection [13,14]. Structural prediction using AlphaFold2 and homology modeling of the NRC3 homodimer revealed that the residues we identified are located on multiple surfaces (Fig 5). These residues are located in an exposed region of the LRR and NB-ARC domains, facing the cavity in

between the LRR and NB-ARC domains, and positioning at the interface between the two NRC3 protomers. Furthermore, upon activation by a compatible sensor NLR, NRC3 migrated as high-molecular-weight complexes with a greatly reduced amount of homodimers (Fig 6C). We propose that these residues may participate in sequential transient interactions between sensor NLRs and NRCs, allowing sensor NLRs to first associate with its matching NRC and then disrupt the interface between the two NRC protomers before the assembly of NRC into activated hexameric resistosomes. Capturing and resolving the structures of these transient sensor-helper NLR resistosome complexes could shed light on the intricate mechanisms governing sensor-helper compatibility and signal transduction.

## Materials and methods

### Plant material and growth conditions

Wild type (WT) and *nrc2/3/4_KO N. benthamiana* [23] were grown in a walk-in chamber with a temperature of 25˚C, humidity of 45–65%, and 16/8 hr light/dark cycle.

### Agroinfiltration

We used *Agrobacterium tumefaciens* to transiently express the NRC3 variants, sensor NLRs, and effectors in four-week-old *N. benthamiana*. Strains of *A. tumefaciens* were refreshed from glycerol stock to 523 medium containing appropriate antibiotics at 28˚C overnight. Cells were harvested by centrifugation at 2500 × g, room temperature for 5 min. Cells were resuspended in MMA buffer (10 mM MgCl2, 10 mM MES-KOH, 150 µM acetosyringone, pH5.6) to the concentration listed in S1 Table and then infiltrated into leaves using 1 mL syringes.

### *Phytophthora infestans* and disease resistance assay

*Phytophthora infestans* 214009 (a gift from Dr. Jin-Hsing Huang, Taiwan Agriculture Research Institute) was isolated from Taiping, Taichung City, Taiwan. *P. infestans* was maintained in the Rye medium at 19˚C in dark conditions [35]. Before the infection assay, the *P. infestans* was subcultured on tomato (cv. money maker) leaf discs. In the disease resistance assay, four-week-old *N. benthamiana* plants were vacuum infiltrated with *A. tumefaciens* containing constructs of interest shown in S2 Table. Sporangia were collected from *P. infestans*-infected tomato leaf discs by gently shaking at 4˚C sterile water. Droplets of 10µL of sporangia suspensions (3 x $10^3$ sporangia/mL) were applied to the abaxial side of detached *N. benthamiana* leaves 6 hours post vacuum infiltration. The infected leaves were placed in the Corning 245 mm Square Dish at 19˚C in a dark condition. The disease symptoms were quantified using UVP ChemStudio Imaging Systems at 5 days post-infection (dpi). Blue LED light was used for excitation and FITC filter (519 nm) and Cy5 filter (670 nm) were used to detect the autofluorescence from infected and healthy tissues.

### Plasmid constructions

All the NRC3 variants used in this study were constructed using Golden Gate assembly [36,37]. The natural variants were amplified from genomic DNA and cloned into a binary vector (pICH86988) which carries the 35S promoter and OCS terminator. The petunia NRC3 was synthesized and then cloned into pICH86988. The auto-active mutants of N88, N89, and NRC3bs were generated through site-directed mutagenesis introducing a D to V mutation in the MHD motif. The chimeric NRC3 variants were made by modularizing regions of NbNRC3 and SlNRC3 as level 0 modules and re-assembled by using Golden Gate assembly. The CC, NB-ARC, and LRR domains were first modularized as level 0 modules to investigate

which domain contributes to the subfunctionalization of NbNRC3c. The stop codon in the LRR domains was removed for tagging with C-terminal tags. These modules were assembled into pICH86988 together with C-terminal 4x myc. To identify the residues that contribute to the subfunctionalization in the NB-ARC domain, the polymorphisms in this domain were introduced into the NbNRC3 NB-ARC module to generate 10 NB-ARC modules. These modules were assembled with CC and LRR domains to obtain full-length NRC3 variants. The polymorphisms in the LRR domain were analyzed using a similar strategy. We first divided this domain into 5 fragments (1 to 5) and made them into independent level 0 modules. We found that the polymorphisms in regions 2 and 5 corporately contributed to the subfunctionalization. Therefore, we further mutated LRR modules of NbNRC3 to introduce the polymorphisms into these regions. These modules were assembled with CC, NB-ARC, and other fragmented LRR modules to become full-length NRC3 variants. All modules, together with C-terminal 4x myc, were assembled into pICH86988. The lysine-asparagine swapping in the N88 and N89 background was done by site-directed mutagenesis. Sequences of plasmids, NRC3 variants, and primers used in this study are listed in S1–S3 Datasets.

## Cell death assay

The complementation assays were conducted in *nrc2/3/4*_KO *N. benthamiana* background, while auto-activity analyses were done in WT *N. benthamiana*. Sensor NLRs and corresponding effectors were transiently expressed by *A. tumefaciens* strains containing expression vectors for different proteins as indicated. The cell death intensity was quantified using UVP Chem-Studio Imaging Systems at 6 days post agroinfiltration (dpi). We used blue LED light as the excitation light and a FITC filter (519 nm) to detect the autofluorescence of necrotic regions. Cell death intensity was normalized by dividing the pixel value by the maximum signal pixel value (65535) detected by the UVP ChemStudio.

## Statistic analysis

Statistical analysis was conducted using R (v4.3.3) [38]. The analyses included Tukey's HSD tests, nonparametric Wilcoxon signed rank tests, or Dunn tests as indicated. The Wilcoxon signed rank tests were used to compare the negative controls (EV, in most cases) or $NN_3NT_{5be}$ (in S14 Fig) with the tested groups. Dunn tests were applied to compare each test group individually, with p-values adjusted using the Bonferroni correction. Thresholds of statistical significance are indicated in the figure legends. Complete statistical data can be found in S7 Data.

## Protein extraction

Leaf tissue was finely powdered by grinding in a mortar and pestle with liquid nitrogen and total protein was subsequently extracted with extraction buffer (10% glycerol, 25 mM Tris pH7.5, 1 mM EDTA, 150 mM NaCl, 2% w/v PVPP, 10 mM DTT, 1x protease inhibitor cocktail (Sigma, P9599), 0.2% IGEPAL (Sigma)). After centrifugation at 13,000 xg for 10 min at 4°C, the supernatant fraction was taken out and mixed with 4x sample loading dye (200 mM Tris-HCl (PH6.8), 8% (w/v) SDS, 40% (v/v) glycerol, 50 mM EDTA, 0.08% bromophenol blue) with 100 mM DTT. Protein samples were incubated at 70°C for 10 minutes before being analyzed by SDS-PAGE.

## SDS-PAGE electrophoresis and immunodetection analyses

SDS-PAGE electrophoresis and immunodetection analyses were conducted as previously described [39,40]. Denatured samples were run on PAGE containing 10% or 15% T-Pro EZ

Gel Solution, 0.1% (v/v) TEMED (Bio-Rad), and 0.1% (w/v) ammonium persulfate (Bio-Rad). The proteins were then transferred to PVDF membranes using the Trans-Blot Turbo Transfer System (Bio-Rad). Immunoblotting and detection were performed on the PVDF membrane with SNAP i.d. 2.0 Protein Detection System (Merck) by using anti-myc (A00704, GenScript), anti-RFP (YH80520, Yao-Hong) and anti-GFP (A-11122, Invitrogen) as primary antibodies and Peroxidase AffiniPure Goat Anti-Mouse IgG (H+L) (115-035-003, Jackson) or Peroxidase Conjugated Goat Anti-Rabbit IgG (H+L) (AP132P, Sigma) as secondary antibodies. First antibodies were at a dilution of 1:8000 while secondary antibodies were at a dilution of 1:25000. The chemiluminescence was detected using SuperSignal West Pico PLUS Chemiluminescent Substrate (34580, Thermo Scientific) mixed with SuperSignal West Femto Maximum Sensitivity Substrate (34096, Thermo Scientific, ratio = 4:1). The images were taken under UVP Chem-Studio Imaging Systems. SimplyBlue SafeStain (465034, Invitrogen) was applied to detect the signal of rubisco on the PVDF membrane.

### Detection of NRC protein accumulation

To detect the protein accumulation of NRC3 variants, four-week-old *N. benthamiana* plants were infiltrated with *A. tumefaciens* containing constructs of NRC3 variants-myc at $OD_{600}$ 0.5. Six leaf discs (1.1 cm diameter) were collected at 2 dpi and protein accumulation was detected according to the immunoblot analysis described above.

### Co-immunoprecipitation assay

To detect the steady-state interaction between Rpi-blb2 and NRC3s, four-week-old *N. benthamiana* plants were infiltrated with *A. tumefaciens* containing constructs of interest shown in S3 Table. Six leaves (2.5g) were collected at 30 hpi and extracted with the extraction buffer described above. Anti-c-myc beads (VF299569, Thermo) or RFP-Trap magnetic agarose (Chromotek) were added to the protein extracts and incubated at 4˚C for 3 h. After incubation, the protein mixtures were washed with wash buffer (10% glycerol, 25 mM Tris pH7.5, 1 mM EDTA, 150 mM NaCl, and 0.2% IGEPAL) five times to remove non-specific binding proteins. To elute proteins, beads or magnetic agarose were mixed with 1x sample loading dye with 100 mM DTT and incubated at 70˚C for 10 minutes before analyzing by SDS-PAGE. Protein signals were detected through the process described above.

### Blue native polyacrylamide gel electrophoresis (BN-PAGE)

To determine the size of NRC3 under native conditions, four-week-old *N. benthamiana* plants were infiltrated with *A. tumefaciens* expressing $N^{L21E}NN/N^{L21E}N_{PKK}N_{THK}$, Rpi-blb2, with or without AVRblb2. Six leaves (2.5 g) were collected at 2 dpi and extracted using native PAGE extraction buffer (50 mM Hepes, pH 7.5, 50 mM NaCl, 5 mM MgCl2, 10% glycerol, 10 mM DTT, 1x protease inhibitor cocktail (Sigma, P9599), and 1% digitonin (Sigma, D141)). The protein extracts were incubated for 10 minutes at 4˚C, then centrifuged at 13,000 xg for 10 minutes at 4˚C. The supernatant was transferred to a new tube, diluted to 0.4X, and mixed with 4× NativePAGE Sample Buffer (Invitrogen, BN2003) and NativePAGE 5% G-250 Sample Additive (Invitrogen, BN2004), resulting in a final G-250 Sample Additive concentration of 0.125%. The electrophoresis were performed using NativePAGE 4–16% Bis-Tris Gel (Invitrogen, BN1002). The proteins were then transferred to PVDF membranes using the Trans-Blot Turbo Transfer System (Bio-Rad) and fixed by incubating in 8% acetic acid for 15 minutes, rinsing with ddH₂O, and allowing to dry. The membranes were reactivated with 99% ethanol, rinsed with ddH₂O and TBST, and subjected to immunodetection analyses as described above.

## Entropy analysis

The sequences of NRC3 of tomato (*Solanum pennellii*, *Solanum chilense*, and *Solanum lycopersicum*), tobacco (*Nicotiana tabacum* and *Nicotiana benthamiana*), pepper (*Capsicum annuum*), eggplant (*Solanum melongena*), and potato (*Solanum tuberosum*) were downloaded from Sol Genomics Network (SGN). The sequences of NRC3 of *Nicotiana sylvestris* were downloaded from NCBI. The accession numbers are listed in S3 Dataset. The protein sequences of NRC3s were aligned using MAFFT (version 7) and used for the entropy analysis. The Shannon entropy was calculated using an online tool established by Los Alamos National Laboratory (https://www.hiv.lanl.gov/content/sequence/HIV/HIVTools.html). A score of 1.5 was set as a cutoff to determine highly variable residues [41]. Detailed results of the entropy analysis are included in S4 Dataset.

## Protein structure prediction

To understand the spatial arrangement of residues in NbNRC3, we used AlphaFold2 to predict the 3D structures of NbNRC3 [42]. This prediction was solely based on the amino acid sequence, without relying on a template. By following guidelines from https://github.com/deepmind/alphafold, we applied AlphaFold2 on ASGC (Academia Sinica Grid Computing Center) to create structural predictions. In the resulting NbNRC3 structure, the region from residue 157 to 888 closely resembled the structure of a corresponding region (residues 152–885) in NbNRC2 (PDB: 8RFH), with an RMSD of 2.155. Assuming similar protein conformation, we predicted the NbNRC3 dimer structure by aligning it with the resting NbNRC2 homodimer structure (PDB: 8RFH) using the Chimera "MatchMaker" function, creating the NbNRC3 homodimer structure (S18 Fig) [19,43]. The N-terminal coiled-coil domain from residues 1 to 152 were not resolved in the NbNRC2 cryo-EM structure (PDB: 8RFH). Therefore, the arrangement of the coiled-coil domain (residues 1–140), linked by a flexible loop (residues 141–160), may differ from our current model (S18 Fig).

## Phylogenetic analysis and detection of selection

Sequences of NRCs from *Capsicum annuum*, *Solanum lycopersicum*, *Nicotiana benthamiana*, and *Solanum tuberosum* were downloaded from the databases in Sol Genomics Network [44]. The full-length nucleotide sequences of NRC or NRC3 variants were translated into protein sequences and aligned by using MEGA software (Molecular Evolutionary Genetics Analysis) [45]. The gaps within alignments were removed manually. The trimmed amino acid sequences were applied to phylogenetic analysis by using Maximum-likelihood phylogenetic analysis with the evolutionary model JTT+G+I and 1000 bootstrap tests. The aligned NRC3 protein sequences were converted into nucleotide sequences for detection of selection. The sequences were analyzed by SLAC (Single-Likelihood Ancestor Counting) analysis and FEL (Fixed Effects Likelihood) analysis on the Datamonkey Adaptive Evolution Server (https://www.datamonkey.org/) [46–48]. Detailed results of the SLAC and FEL analyses are included in S5 Dataset.

## Cell biology assays

To detect the punctate formation of NRCs in response to immune activation, four-week-old *N. benthamiana* were agroinfiltrated with constructs of interest shown in S4 Table. Leaf tissues were imaged at 3 dpi by using Olympus FV3000 confocal microscope with a 60x silicone oil immersion objective with the excitation of 488 nm for GFP and 561 nm for mCherry. The emission wavelengths for GFP and mCherry tags were set to 500-530 nm and 590-620 nm, respectively.

## Ancestral sequence reconstruction

Sequences of the NRC family members were extracted from 124 Solanaceae genomes using NLRtracker followed by a phylogeny-based approach [49,50]. To identify the NRCX/1/2/3 clades, we aligned the 1,116 protein sequences of the NB-ARC domain of the NRC family using MAFFT v7.508 with default options. Sequences of NB-ARC regions extracted in Adachi et al., 2023 were also included in the analysis [25]. We then reconstructed a phylogenetic tree using FastTree v2.1.11 [51] and extracted 324 NRC sequences within the NRCX/1/2/3 clades. To obtain a codon-based nucleotide sequence alignment, we re-aligned 324 full-length protein sequences of the NRCX/1/2/3 clades using MAFFT with default options and, then, we threaded the nucleotide sequences onto the protein alignment using the "thread_dna" command in phykit v1.11.14 [52,53]. Based on this nucleotide sequence alignment, we reconstructed the phylogenetic tree of NRCX/1/2/3 using IQ-TREE v2.2.0.3 (http://www.iqtree.org) [54] with 1,000 ultrafast bootstrap replicates [55]. The best-fit model to reconstruct the tree was automatically selected by ModelFinder [56] in IQ-TREE, and the "TIM3+F+I+I+R4" model has been selected according to the Bayesian Information Criterion (BIC). Based on this phylogenetic tree and codon-based alignment, we reconstructed ancestral sequences using FastML with the nucleotide sequence mode and its default settings (S6 Dataset). The five selected NRC3 ancestral variants were synthesized using the service provided by SynBio Technologies (New Jersey, USA) and then cloned into pICH86988.

## Supporting information

**S1 Fig. The NRC from solanaceous plants were grouped into NRC0 to NRCX based on the phylogenetic analysis.** Phylogenetic tree of NRC from tomato, tobacco, potato, and pepper. The sequences of NRCs were downloaded from the Sol Genomics Network. The alignment of amino acid sequences of the NB-ARC domain was used for phylogenetic analysis using the Maximum-likelihood method with 1000 bootstrap tests. Gray boxes indicated different NRC clades. The clade of NRC0 was used as an outgroup. Tomato and *N. benthamiana* NRCs used in Figs 1A and S2 were highlighted in red and green, respectively.
(JPF)

**S2 Fig. NRCs from solanaceous plants displayed diverse compatibility with sensor NLRs.** (A) Workflow of cell death assays testing the ability of NRC to function with different sensor NLRs. NRCs were co-expressed with sensor NLRs and the corresponding effectors in *nrc2/3/4_KO N. benthamiana*. NRC variants were co-expressed with (B) Rx/CP, (C) Sw5b/NSm, (D) Gpa2/RBP1, (E) Pto/AvrPto, or (F) R1/AVR1 to analyze their ability to work with sensor NLRs. Cell death phenotypes were recorded at 6 dpi. The dot plots represent cell death intensity quantified using UVP ChemStudio PLUS. The line in the boxplots represents the medium, the box edges represent the 25th and 75th percentiles, and the whiskers extend to the most extreme data points no more than 1.5x of the interquartile range. Statistical differences between the negative control (EV) and tested groups were examined by paired Wilcoxon signed rank test (* = p < 0.0001).
(JPF)

**S3 Fig. NRC3 natural variants show no or low auto-activities when expressed in *N. benthamiana*.** Auto-activity analysis of NRC3 natural variants tested in Fig 1D. The NRC3 variants were expressed alone in WT *N. benthamiana*. The dot plots represent cell death intensity quantified using UVP ChemStudio PLUS at 6 dpi.
(JPF)

**S4 Fig. Ancestral sequence reconstruction of NRC3.** (A) Phylogenetic tree of the solanaceous NRC3 clade. The NLR sequences were extracted from 124 genomes in the Solanaceae family using the NLRtracker software. The phylogenetic tree was reconstructed based on the codon-based nucleotide sequence alignment of NRCX, NRC1, NRC2, and NRC3 clades using IQ-TREE. Ancestral sequence reconstruction was performed using FastML. The nodes of ancestral variants tested in this study were indicated with star shapes. N3: the ancestral variant before the divergence of the three allelic groups; N4: the ancestral variant of NRC3a; N88: the ancestral variant before the divergence of NRC3b/c; N95: the ancestral variant of NRC3b; N89: the ancestral variant of NRC3c. (B) Polymorphisms at the six positions identified across NRC3 natural variants.
(JPF)

**S5 Fig. Amino acid sequence alignment of NRC3 ancestral variants and natural variants.** The alignment was done using MAFFT (Multiple Alignment using Fast Fourier Transform). N3: the ancestral variant before the divergent of three allelic groups. N4: the ancestral variant of NRC3a. N88: the ancestral variant before divergent of NRC3b/c. N95: the ancestral variant of NRC3b. N89: the ancestral variant of NRC3c.
(JPF)

**S6 Fig. NRC3 ancestral variants show no or low auto-activities when expressed in *N. benthamiana*.** (A) Auto-activity analysis of ancestral NRC3 variants. The NRC3 variants were expressed alone in WT *N. benthamiana*. Cell death intensity and phenotypes were recorded at 6 dpi. The dot plots represent cell death intensity quantified using UVP ChemStudio PLUS. (B) Protein accumulation of ancestral NRC3 variants. NRC3 variants were transiently expressed in WT *N. benthamiana*. The proteins were extracted from leaf samples at 2 dpi and the NRC3 protein accumulations were detected by α-myc antibody. SimplyBlue SafeStain-staining of Rubisco was used as the loading control.
(JPF)

**S7 Fig. The manually curated PinNRC3 variant rescued Rpi-blb2/Prf/Rx-mediated cell death.** (A) Amino acid sequence comparison of SlNRC3, *Petunia* NRC3 from the genome database and curated version. The indel was manually curated using sequences from SlNRC3. (B) Cell death assays of PinNRC3 variants co-expressed with Rpi-blb2/AVRblb2, Pto/AvrPto, or Rx/CP in *nrc2/3/4_KO N. benthamiana*. The auto-activity analysis was done by expressing NRC3 variants alone in WT *N. benthamiana*. The dot plots represent cell death intensity quantified by UVP ChemStudio PLUS at 6 dpi. The line in the boxplots represents the medium, the box edges represent the 25th and 75th percentiles, and the whiskers extend to the most extreme data points no more than 1.5x of the interquartile range. Statistical differences between the negative control (EV) and tested groups were examined by paired Wilcoxon signed rank test (* = $p < 0.05$).
(JPF)

**S8 Fig. The NB-ARC and LRR domains cooperatively contributed to the sensor-helper compatibility.** (A) Cell death assays of chimeric NbNRC3/SlNRC3 variants. The NRC3 chimeric variants were co-expressed with Rpi-blb2/AVRblb2 or Pto/AvrPto in *nrc2/3/4_KO N. benthamiana*. Auto-activity analysis was done by expressing NRC3 variants alone in WT *N. benthamiana*. The dot plots represent cell death intensity quantified by UVP ChemStudio PLUS at 6 dpi. The line in the boxplots represents the medium, the box edges represent the 25th and 75th percentiles, and the whiskers extend to the most extreme data points no more than 1.5x of the interquartile range. Statistical differences were examined using Dunn's test ($p < 0.05$). (B) Protein accumulation of chimeric NRC3 variants tested in (A). NRC3 variants

were transiently expressed in WT *N. benthamiana*. The proteins were extracted from leaf samples at 2 dpi and the NRC3 protein accumulations were detected by α-myc antibody. SimplyBlue SafeStain-staining of Rubisco was used as the loading control.
(JPF)

**S9 Fig. Region 3 of the NB-ARC domain contributed to the sensor-helper compatibility.**
(A) Cell death assays of chimeric NRC3 variants designed for investigating regions of the NB-ARC domain that contribute to sensor-helper compatibility. The variants were co-expressed with Rpi-blb2/AVRblb2 or Pto/AvrPto in *nrc2/3/4_KO N. benthamiana*. Auto-activity analysis was done by expressing NRC3 variants alone in WT *N. benthamiana*. The dot plots represent cell death intensity quantified by UVP ChemStudio PLUS at 6 dpi. The line in the boxplots represents the medium, the box edges represent the 25th and 75th percentiles, and the whiskers extend to the most extreme data points no more than 1.5x of the interquartile range. Statistical differences were examined using Dunn's test (p < 0.05). (B) Protein accumulation of chimeric NRC3 variants tested in (A). NRC3 variants were transiently expressed in WT *N. benthamiana*. The proteins were extracted from leaf samples at 2 dpi and the NRC3 protein accumulations were detected by α-myc antibody. SimplyBlue SafeStain-staining of Rubisco was used as the loading control.
(JPF)

**S10 Fig. NRC3 variants NSN$_2$ to NSN$_5$ and NSN$_{25}$ functioned with Prf.** (A) Cell death assays of chimeric NRC3 variants tested in Fig 3C. The variants were co-expressed with Pto/AvrPto in *nrc2/3/4_KO N. benthamiana*. Auto-activity analysis was done by expressing NRC3 variants alone in WT *N. benthamiana*. The dot plots represent cell death intensity quantified by UVP ChemStudio PLUS at 6 dpi. The line in the boxplots represents the medium, the box edges represent the 25th and 75th percentiles, and the whiskers extend to the most extreme data points no more than 1.5x of the interquartile range. Statistical differences were examined using Dunn's test (p < 0.05). (B) Protein accumulation of chimeric NRC3 variants tested in (A). NRC3 variants were transiently expressed in WT *N. benthamiana*. The proteins were extracted from leaf samples at 2 dpi and the NRC3 protein accumulations were detected by α-myc antibody. SimplyBlue SafeStain-staining of Rubisco was used as the loading control.
(JPF)

**S11 Fig. A single amino acid change (I642T) conferred full activity in rescuing Rpi-blb2 cell death in the NSN$_5$ background.** (A) Cell death assays of chimeric NRC3 variants carrying T607A or I642T in NSN background. The variants were co-expressed with Rpi-blb2/AVRblb2 or Pto/AvrPto in *nrc2/3/4_KO N. benthamiana*. (B) Cell death assays of chimeric NRC3 variants tested in Fig 3D. The variants were co-expressed with Pto/AvrPto in *nrc2/3/4_KO N. benthamiana*. Auto-activity analysis was done by expressing NRC3 variants alone in WT *N. benthamiana*. The dot plots represent cell death intensity quantified by UVP ChemStudio PLUS at 6 dpi. The line in the boxplots represents the medium, the box edges represent the 25th and 75th percentiles, and the whiskers extend to the most extreme data points no more than 1.5x of the interquartile range. Statistical differences were examined using Dunn's test (p < 0.05). (C and D) Protein accumulation of chimeric NRC3 variants tested in (A) and (B). NRC3 variants were transiently expressed in WT *N. benthamiana*. The proteins were extracted from leaf samples at 2 dpi and the NRC3 protein accumulations were detected by α-myc antibody. SimplyBlue SafeStain-staining of Rubisco was used as the loading control.
(JPF)

**S12 Fig. The NRC3 variant NSN$_{25be}$ fully rescued the Rpi-blb2-mediated cell death in the *nrc2/3/4_KO N. benthamiana*.** (A) Cell death assays of chimeric NRC3 variants carrying LRR

region 5a to 5e of SlNRC3 in NSN background. The variants were co-expressed with Rpi-blb2/AVRblb2 or Pto/AvrPto in *nrc2/3/4*_KO *N. benthamiana*. (B) Cell death assays of chimeric NRC3 variants tested in Fig 3E. The variants were co-expressed with Pto/AvrPto in *nrc2/3/4*_KO *N. benthamiana*. Auto-activity analysis was done by expressing NRC3 variants alone in WT *N. benthamiana*. The dot plots represent cell death intensity quantified by UVP ChemStudio PLUS at 6 dpi. The line in the boxplots represents the medium, the box edges represent the 25th and 75th percentiles, and the whiskers extend to the most extreme data points no more than 1.5x of the interquartile range. Statistical differences were examined using Dunn's test (p < 0.05). (C and D) Protein accumulation of chimeric NRC3 variants tested in (A) and (B). NRC3 variants were transiently expressed in WT *N. benthamiana*. The proteins were extracted from leaf samples at 2 dpi and the NRC3 protein accumulations were detected by α-myc antibody. SimplyBlue SafeStain-staining of Rubisco was used as the loading control. (JPF)

**S13 Fig. NN$_3$N$_{T5be}$ functioned with Prf and was not auto-active.** (A) Cell death assays of chimeric NRC3 variants tested in Fig 3F. The variants were co-expressed with Pto/AvrPto in *nrc2/3/4*_KO *N. benthamiana*. Auto-activity analysis was done by expressing NRC3 variants alone in WT *N. benthamiana*. The dot plots represent cell death intensity quantified by UVP ChemStudio PLUS at 6 dpi. The line in the boxplots represents the medium, the box edges represent the 25th and 75th percentiles, and the whiskers extend to the most extreme data points no more than 1.5x of the interquartile range. Statistical differences were examined using Dunn's test (p < 0.05). (B) Protein accumulation of chimeric NRC3 variants tested in (A). NRC3 variants were transiently expressed in WT *N. benthamiana*. The proteins were extracted from leaf samples at 2 dpi and the NRC3 protein accumulations were detected by α-myc antibody. SimplyBlue SafeStain-staining of Rubisco was used as the loading control. (JPF)

**S14 Fig. NRC3 variants with S202P, T203K, N221K, C824H, N832K, or V881I mutations quantitatively affected the ability of NRC3 to function with Rpi-blb2.** Cell death assays of chimeric NRC3 variants designed for pinpointing the polymorphisms in (A) NB-ARC domain region 3 which contains six amino acid differences, (B) LRR domain region 5b containing five amino acid differences, and (C) LRR domain region 5e contain a small insertion/indel (labeled as X) three amino acid (TIH) differences between NbNRC3c and SlNRC3a. The analysis was done in the NN$_3$N$_{T5be}$ background by replacing each residue with the amino acid of NbNRC3c. The variants were co-expressed with Rpi-blb2/AVRblb2 or Pto/AvrPto in *nrc2/3/4*_KO *N. benthamiana*. Mutations at positions 202, 203, 221 (NB-ARC domain), 824, 832 (LRR domain region 5b), and 881 (LRR domain region 5e) affected the ability of NN$_3$N$_{T5be}$ to function with Rpi-blb2. Auto-activity analysis was done by expressing NRC3 variants alone in WT *N. benthamiana*. The dot plots represent cell death intensity quantified by UVP ChemStudio PLUS at 6 dpi. The line in the boxplots represents the medium, the box edges represent the 25th and 75th percentiles, and the whiskers extend to the most extreme data points no more than 1.5x of the interquartile range. Statistical differences between the NN$_3$N$_{T5be}$ and tested groups were examined by paired Wilcoxon signed rank test (* = p < 0.01, ** = p < 0.001, *** = p < 0.0001). (D-F) Protein accumulation of chimeric NRC3 variants tested in (A-C). The proteins were extracted from leaf samples at 2 dpi and the NRC3 protein accumulations were detected by α-myc antibody. SimplyBlue SafeStain-staining of Rubisco was used as the loading control. (JPF)

**S15 Fig. NNPKKNTHKI and NNPKKNTHK functioned with Prf and were not auto-active.** (A) Cell death assays of chimeric NRC3 variants tested in Fig 3H. The variants were co-

expressed with Pto/AvrPto in *nrc2/3/4_KO N. benthamiana*. Auto-activity analysis was done by expressing NRC3 variants alone in WT *N. benthamiana*. The dot plots represent cell death intensity quantified by UVP ChemStudio PLUS at 6 dpi. The line in the boxplots represents the medium, the box edges represent the 25th and 75th percentiles, and the whiskers extend to the most extreme data points no more than 1.5x of the interquartile range. Statistical differences were examined using Dunn's test ($p < 0.05$). (B) Protein accumulation of chimeric NRC3 variants tested in (A). The proteins were extracted from leaf samples at 2 dpi and the NRC3 protein accumulations were detected by α-myc antibody. SimplyBlue SafeStain-staining of Rubisco was used as the loading control.
(JPF)

**S16 Fig. NbNRC3c variants carrying polymorphisms from the natural variants functioned with Prf and were not auto-active.** (A) Left panel, the polymorphisms in natural NRC3 variants at the six positions identified. Right panels, cell death assays testing these polymorphisms in NbNRC3c background. All the variants functioned with Prf and none of them were auto-active. The variants were co-expressed with Pto/AvrPto in *nrc2/3/4_KO N. benthamiana*. Auto-activity analysis was done by expressing NRC3 variants alone in WT *N. benthamiana*. The dot plots represent cell death intensity quantified by UVP ChemStudio PLUS at 6 dpi. The line in the boxplots represents the medium, the box edges represent the 25th and 75th percentiles, and the whiskers extend to the most extreme data points no more than 1.5x of the interquartile range. Statistical differences were examined using Dunn's test ($p < 0.05$). (B) Protein accumulation of NRC3 variants tested in (A). The proteins were extracted from leaf samples at 2 dpi and the NRC3 protein accumulations were detected by α-myc antibody. SimplyBlue SafeStain-staining of Rubisco was used as the loading control.
(JPF)

**S17 Fig. Swapping two K/N residues in N88/N89 do not affect their compatibility with Prf.** (A) Left panel, the polymorphisms in ancestral NRC3 variants at the 6 positions identified. Right panels, cell death assays testing two lysine-asparagine changes in both N88 and N89 backgrounds. All the variants functioned with Prf and none of them were auto-active. The variants were co-expressed with Pto/AvrPto in *nrc2/3/4_KO N. benthamiana*. Auto-activity analysis was done by expressing NRC3 variants alone in WT *N. benthamiana*. The dot plots represent cell death intensity quantified by UVP ChemStudio PLUS at 6 dpi. The line in the boxplots represents the medium, the box edges represent the 25th and 75th percentiles, and the whiskers extend to the most extreme data points no more than 1.5x of the interquartile range. Statistical differences were examined using Dunn's test ($p < 0.05$). (B) Protein accumulation of NRC3 variants tested in (A). The proteins were extracted from leaf samples at 2 dpi and the NRC3 protein accumulations were detected by α-myc antibody. SimplyBlue SafeStain-staining of Rubisco was used as the loading control.
(JPF)

**S18 Fig. Superimposition of predicted NbNRC3c homodimer on NbNRC2 homodimer.** Structure alignment between NbNRC2 and NbNRC3c homodimer shown in two orthogonal views. The remodeled NbNRC3c homodimer includes predicted CC domains. The 3D structure of NbNRC3c was first predicted using AlphaFold2 and aligned to the resting NbNRC2 homodimer structure.
(JPF)

**S19 Fig. NRC3 variants carrying residues from SlNRC3 in different surfaces functioned with Prf and were not auto-active.** (A) All the variants functioned with Prf and none of them were auto-active. The variants were co-expressed with Pto/AvrPto in *nrc2/3/4_KO N.*

*benthamiana.* Auto-activity analysis was done by expressing NRC3 variants alone in WT *N. benthamiana.* The dot plots represent cell death intensity quantified by UVP ChemStudio PLUS at 6 dpi. The line in the boxplots represents the medium, the box edges represent the 25th and 75th percentiles, and the whiskers extend to the most extreme data points no more than 1.5x of the interquartile range. Statistical differences were examined using Dunn's test (p < 0.05). (B) Protein accumulation of NRC3 variants tested in (A). The proteins were extracted from leaf samples at 2 dpi and the NRC3 protein accumulations were detected by α-myc antibody. SimplyBlue SafeStain-staining of Rubisco was used as the loading control. (JPF)

**S20 Fig. Steady-state interactions detected using co-IP did not reflect the compatibility between Rpi-blb2 and NRC3 variants.** Co-immunoprecipitation assays of Rpi-blb2 and NRC3 variants. (A) RFP-Rpi-blb2 and NRC3s-myc were co-expressed with or without Flag-AVRblb2. NRC3s-myc co-expressed with RFP were used as negative controls. Protein extracts (input) and RFP-Trap pull-down samples (IP) were analyzed using western blot analysis with α-RFP, and α-myc antibodies. SimplyBlue SafeStain-staining of Rubisco was used as the loading control. (B) Protein extracts (input) and myc-Trap pull-down samples (IP) were analyzed using western blot analysis using α-RFP, and α-myc antibodies. SimplyBlue SafeStain-staining of Rubisco was used as the loading control. (JPF)

**S21 Fig. NRC3 variants do not form membrane-associated punctate dots in the presence of sensor NLRs without corresponding effectors.** (A) NRCs-GFP were co-expressed with HF-Rpi-blb2 or Pto. Samples were examined at 3 dpi. Scale bars represent 10μm. SOBIR1-m-Cherry was used as a plasma membrane marker. (B) Quantification of punctate dots of NRCs in (A). (JPF)

**S22 Fig. Functional divergence and gene loss of NRCs led to the division of the NLR network into smaller subnetworks.** (A) Evolutionary history of solanaceous NRC3. The ancestral NRC3 (ancNRC3) was first duplicated into two alleles, NRC3a in *Capsicum* and *Solanum* species and ancestral NRC3b/c in *Nicotiana* species. The ancestral NRC3b/c was further duplicated to generate NRC3b and NRC3c. NRC3b later underwent nonfunctionalization and was lost in *N. benthamiana* while NRC3c underwent subfunctionalization. As a result, the NRC3 in *Nicotiana* species lost the ability to function with Rpi-blb2. (B) Evolution of the complex NRC network in plants. The original NLR pair expanded to form an initial NRC network with an ancestral NRC (brown), functioning with multiple sensor NLRs. This ancestral variant duplicated, resulting in two functionally redundant variants: 'b' and 'c' (green and blue, respectively). Variant 'b' lost its sensor NLR compatibility due to gene loss or nonfunctionalization, while variant 'c' retained partial compatibility, undergoing subfunctionalization. Consequently, variant 'c' functions with a narrower range of sensor NLRs, forming a distinct NRC subnetwork. Similar events likely occurred to various NRC variants, not limiting to NRC3 dissected in this study. (JPF)

**S1 Table. List of constructs used in cell death assays.** (PDF)

**S2 Table. List of constructs used in disease resistance assays.** (PDF)

**S3 Table. List of constructs used in co-IP assays.**
(PDF)

**S4 Table. List of constructs used in cell biology assays.**
(PDF)

**S1 Dataset. List of primers used in this study.**
(XLSX)

**S2 Dataset. List of plasmids used in this study.**
(XLSX)

**S3 Dataset. Sequences and accession numbers of NRC used in this study.**
(XLSX)

**S4 Dataset. Results of the entropy analysis.**
(XLSX)

**S5 Dataset. Results of FEL (Fixed Effects Likelihood) analysis and SLAC (Single-Likelihood Ancestor Counting) analysis.**
(XLSX)

**S6 Dataset. Results of ancestral sequence reconstruction.**
(XLSX)

**S7 Dataset. Summary of statistical analysis.**
(XLSX)

## Acknowledgments

We thank Dr. Sophien Kamoun (The Sainsbury Laboratory, UK) for suggestions about the experiments and manuscript, and Dr. Muniyandi Selvaraj (The Sainsbury Laboratory, UK) and Dr. Mauricio Contreras (The Sainsbury Laboratory, UK) for sharing the structure information of NRC2. We thank Mark Youles (SynBio, The Sainsbury Laboratory, UK) for sharing plasmids for molecular cloning. We thank Mei-Jane Fang and Ji-Ying Huang in the Live-Cell-Imaging Core Lab (Institute of Plant and Microbial Biology, Academia Sinica, Taipei, Taiwan) for help with confocal imaging. We thank Dr. Jin-Hsing Huang (Taiwan Agriculture Research Institute) for providing the *P. infestans* isolated 214009 and advice on pathogen inoculation. The AlphaFold2 prediction utilized the resources of ASGC (Academia Sinica Grid-Computing Center) Distributed Cloud, supported by Academia Sinica.

## Author Contributions

**Conceptualization:** Ching-Yi Huang, Yu-Seng Huang, Lida Derevnina, Chih-Hang Wu.

**Data curation:** Ching-Yi Huang, Yu-Seng Huang, Yu Sugihara, Lo-Ting Huang, AmirAli Toghani, Jiorgos Kourelis, Chih-Hang Wu.

**Formal analysis:** Ching-Yi Huang, Yu-Seng Huang, Chun-Hsiung Wang, Chih-Hang Wu.

**Funding acquisition:** Chih-Hang Wu.

**Investigation:** Ching-Yi Huang, Yu-Seng Huang, Yu Sugihara, Lida Derevnina, Chih-Hang Wu.

**Methodology:** Ching-Yi Huang, Yu-Seng Huang, Yu Sugihara, Hung-Yu Wang, Yi-Feng Chen, Kuan-Yu Lin, Bing-Jen Chiang, Chun-Hsiung Wang, Chih-Hang Wu.

**Project administration:** Chih-Hang Wu.

**Resources:** Ching-Yi Huang, Yu-Seng Huang, Juan Carlos Lopez-Agudelo, Chih-Hang Wu.

**Supervision:** Chih-Hang Wu.

**Validation:** Ching-Yi Huang, Yu-Seng Huang, Chih-Hang Wu.

**Visualization:** Ching-Yi Huang, Yu-Seng Huang, Chih-Hang Wu.

**Writing – original draft:** Ching-Yi Huang, Yu-Seng Huang, Lida Derevnina, Chih-Hang Wu.

**Writing – review & editing:** Ching-Yi Huang, Yu-Seng Huang, Yu Sugihara, Juan Carlos Lopez-Agudelo, Yi-Feng Chen, Jiorgos Kourelis, Lida Derevnina, Chih-Hang Wu.

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
