## [Decision Letter · Decision Letter 0]

2 Apr 2024

Dear Dr Wu,

Thank you very much for submitting your Research Article entitled 'Functional divergence shaped the network architecture of plant immune receptors' to PLOS Genetics.

The manuscript was fully evaluated at the editorial level and by independent peer reviewers. The reviewers appreciated the attention to an important problem, but raised some substantial concerns about the current manuscript. Based on the reviews, we will not be able to accept this version of the manuscript, but we would be willing to review a much-revised version. We cannot, of course, promise publication at that time.

If you decide to revise the manuscript for further consideration at PLOS Genetics, please aim to resubmit within the next 60 days, unless it will take extra time to address the concerns of the reviewers, in which case we would appreciate an expected resubmission date by email to plosgenetics@plos.org.

We are sorry that we cannot be more positive about your manuscript at this stage. Please do not hesitate to contact us if you have any concerns or questions.

Yours sincerely,

Tiancong Qi

Academic Editor

PLOS Genetics

Tanja Slotte

Section Editor

PLOS Genetics

Reviewer's Responses to Questions

**Comments to the Authors:**

Reviewer #1: Review of manuscript PGENETICS-D-24-00215

Sensor NLRs, effector-detecting NLRs, in solanaceous plants rely on the presence and function of helper NLRs of the NRC family. This dependency evolved into a highly expanded NRC-network in Solanaceae and Convolvulaceae. In this manuscript the authors wanted to understand the molecular mechanism/basis for the evolution of this complex network architecture. They picked out one NRC helper NLR, NRC3, and studied the functional divergence of various NRC3 variants. The authors phylogenetic analysis provide evidence that NRC3s form three allelic groups with different specificities/compatibilities for sensor NLRs, mainly Rpl-blb2. By the employment of ancestral sequence reconstruction and analysis of natural and chimeric variants the authors identified six (!) amino acids being key in the sensor-helper compatibility in the helper NRC3. These residues lie in the NB-ARC and LRR domain. The authors hypothesize that these residues are important for the difference in a supposed transient interaction (pre- and post-effector sensing) of the sensor with the helper, having a ‘stronger or longer’ transient interaction between compatible sensor-helper NLR variants. This is based on a TurboID proximity labeling experiment. The authors further show that only compatible sensor-helper combinations lead to the formation of helper NLR puncta at the membrane and thus resistosome formation.

In their final model the authors suggest that the helper NLR, NRC3, homodimer is ‘targeted’ by the effector-modified/bound compatible sensor and transiently interacts with the identified residues, which are located at multiple surfaces of the helper NLR. These transient interactions lead to the disruption of the helper homodimer and the formation of the helper sextuple resistosome.

The paper is very well written, and the experiment are presented in a very appealing manner and are of great quality!

The conclusions drawn by the authors make for sure sense, however, I am not so convinced by the difference in the transient interaction of the compatible vs. the non-compatible helper with the Rlp-blb2 sensor.

In Figure 5 D the one lane is their evidence that the transient interaction of compatible NRC3 is stronger/longer upon AVRblb2 expression than the non-compatible, because the increase in biotin labeling is not seen for the incompatible NRC3. I think this is weak evidence. Another closer time point would maybe help to show this enhanced biotinylation visible in more than one lane/sample.

Another explanation would be that the (transient) interaction strength is not the cause of the incompatibility, but that the effector-bound/modified sensor-NLR is unable to break the homodimer in the incompatible situation, right. I think this possibility is not excluding the ‘interaction-strength’ hypothesis, and the mechanism could rather be a combination of both. The sensor NLR, upon binding/detection of the effector changed confirmation so that now the interaction with the helper, which also happened pre-effector recognition, can break the homodimer via interaction with the specific herein identified residues, which than leads to NRC activation. The six residues that the authors identified in the NRC3 protein are not required for sensor-helper steady-state interaction, but for the activation process. Thus, the transient interaction – at least in my opinion, is the same for both compatible and incompatible helper NRC3 – for sure pre-effector recognition.

Minor issues:

Title: functional divergence – is the paper really providing evidence for functional divergence in the meaning of new function or diverged functions of the NRCs? Or is it rather a functional specification in regard to what sensor NLRs can use which NRC? I am bringing this up as a non-evolutionary biologist, because for me functional divergence was rather associated with a neo-functionalization. Well, as I said, this way of thinking of it may be wrong and I would be happy to be corrected and to be able to learn something new.

Line 52: Is it really shown for plant NLRs that they hydrolyze ATP? Or is it rather so that they can bind both ADP and ATP/dATP?

Lines 197ff: To which NRC3 clade (a, b or c) does PinNRC3 belong to?

Figure 5:

Could the authors do the same experiment in presence of protease inihibitors? Maybe this would also help to see a potential difference in biotinylation even better. Maybe the time points chosen could also be put closer together, lets say 5, 6, 8 and 10 hours?

Why is the biotin labeled Rpi-blb2 not accumulating over time? Is this due to rapid degradation of Rpi-blb2? The potential rapid degradation of Rpi-blb2 makes it difficult to conclude much.

Is the biotin provided enough to also show biotinylation of Rpi-blb2 at the later time points. A least for the incompatible NRC3 it should be the same over time, no? Is it possible to co-express NRC3 with another tag to show biotinylation of this NRC3 (in the homodimer and/or the resistosome) in the samples?

Material and methods:

Cell death quantification – It would be nice to indicate what region of the infiltrated region was used for quantification. What is exactly meant by “Cell death intensity was normalized by dividing the pixel value by the maximum pixel value of the signal (65535).” Is (65535) regarding to a reference?

Reviewer #2: (The review is not uploaded as an attachment.)

In the manuscript, Huang and Huang et al. provide a compelling analysis of both natural and chimeric variants of NRC3, highlighting crucial amino acids involved in sensor-helper compatibility and offering insights into the evolutionary dynamics of the NLR network. Despite the strengths of the study, I believe its impact could be further enhanced by addressing certain areas through additional experiments and discussions.

Major Concerns:

1. The use of AlphaFold2 for predicting the structure of the NRC3 homodimer is acknowledged. However, the absence of experimental validation to confirm NRC3's dimerization in its resting state is a significant oversight. Given the complex nature of protein interactions and the crucial role of oligomer states in protein functionality, integrating experimental evidence for the predicted dimerization would substantiate the findings. Employing Blue Native PAGE analysis, similar to the approach for NRC2 homodimer studies, could offer the needed validation, thereby solidifying the structural assumptions presented.

2. The manuscript intriguingly marks key natural mutations on the predicted model in Figure 4, leading to a section rich in predictions and hypotheses. I recommend relocating this part to the discussion section of the manuscript. This move would not only clarify the distinction between the results and speculative interpretations but also provide a more appropriate context for exploring the implications of these findings.

3. The innovative application of TurboID for probing transient protein interactions is commendable. Yet, the interpretation of the labeling results warrants further scrutiny. The reported pattern of a strong biotin-labeled signal at 6 hours post-treatment, diminishing at 12 hours and disappearing by 30 hours, calls for a deeper exploration into the stability and fate of the proteins initially detected. A discussion on potential rapid degradation or other processes affecting the observed signal dynamics would be valuable. Additionally, referencing the initial TurboID application in plants (PMID:31535972), which showed progressively stronger signals from 15 minutes to 3 hours post-treatment, could provide a benchmark for these results. Such a comparison might illuminate whether the differences observed are due to experimental design or specific protein behaviors, offering a richer interpretation of the data.

Reviewer #3: Ching-Yi Huang and colleagues present a body of work investigating the determinants responsible for functional divergence conditioning signaling specificity in a helper-sensor NLR network constituting the plant immune system in Solanaceae plants. This work conceptually makes an incremental contribution to our understanding on how NLR network has evolved to modulate signaling, initiated from hypervariable NLR sensors, funneling through a group of helper NLRs. In the past, the expansion of NRCs in the clade of Solanaceae had been studied in detail to match the signaling partners of handful sensor NLRs, covering a wide range of pathogens as entry points. This work focused on the compatibility of one of the NRC groups, NRC3 clade, and its cognate pair sensor NLR, Rpi-blb2, since the signaling partnership of this sensor-helper pair is on and off across the Solanaceae species, for example in tomato and tobacco lineages. The level of execution in their investigation of the compatibility of sensor-helper NLR is quite exquisite; the authors were able to 1) narrow down the causal residues contributing to the specificity down to six via an extensive series of domain-swapping followed by mutation analyses and quantitation of HR assays in N. benthamiana, 2) succeeded to experimentally address ancestral function in regards to the compatibility using the synthetic, ancestral NRCs, 3) provide molecular explanation on the time course scenarios of helper-dimer activation towards high-order oligomers, inferred from Turbo-ID experiments and cell biological assessment of the punctate formation on the membrane. The series of data provided convinces this reviewer of the mode of action of NRC3 variants, supporting the notion of functional divergence of NRC3 in splitting a subnetwork in terms of fine-tuning sensor choices.

While the breadth of data is quite amazing and complete with room for minor improvements, the authors might need to spend some other time to scaffold their findings in the context of whole NRC network, if the authors’ main point is to talk about network architecture. In other words, the current presentation and writing shall improve to clarify academic novelty of this work. As alluded in the beginning of this report (with a comment on incremental finding), the authors are encouraged to restructure the introduction to compare this finding to the already published NRC functional divergence. Elaboration on the NRC3 lineage presented in this work is well appreciated; however, the title and overall discussion points on functional divergence of NRCs might not sound new to fill the gap of knowledge. There must be an excitement of the new knowledge built from the currently presented new dataset, which is at the moment difficult to find. If the authors want to highlight the mode of operation of NRC3 variants as a new finding with discovering how functional divergence can be made over evolutionary time-scale, the focus of the manuscript shall be repackaged to convey the strength of this beautiful, extensive work.

With the above-mentioned notion, detailed comments are provided below.

• The title does not fully convey the main findings in the manuscript but remains quite vague and overstretched. The network is not even presented as a main figure either for readers to assess how functional divergence affected the “architecture” of network. Even with the sup figure, I would not call this as an architecture. This is just one layer of a network, being separated by sub-functionalization, which is quite common. To avoid a confusion that it might cause in systems biology, title revision is highly recommended. The current title might not be the strongest point of this work, given that functional divergence of sub-NRC clades were already reported extensively.

• Issues in regards to Turbo-ID experiment

Induction of the sensor NLR expression assessed by western blot peaked at 12hours post induction as shown in Fig 5B. However, the IPed pool of Rpi-blb2 at 6 hours vs. 12 hours in the panel D is quite different with much more intensity found in the 6-hour samples. I wonder whether the faint detection of Strep-decorated sensor NLRs in 12-hour panel would be due to the less efficient IP condition. I wonder the claim of this “transient” interaction based on this data quality could be justified. Could it be just associated with rapid turn-over of the sensor NLR?

Another notion is that biotin-treatment was carried out at one point, namely at the induction point, and once a protein is biotinylated, it remains as biotinylated. To really address “transient” nature of the proximity labeling, the same duration shall be given between the biotin-labeling and sampling time.

In addition, L370-377: description shall be better made to explain whether or not the Streptavidin-decorated sensor NLR is from the IPed sample.

• A scaffolding of knowledge and pre-existing information shall be better addressed. Here I pinpoint several places where changes would be beneficial.

1) Introduction of the functional diversification of NRCs across different lineages shall be better explained starting from Fig. S1. NRCs themselves have undergone duplication events followed by functional diversification. How much is known about the ancestral function basal to NRC1/2/3/X vs. NRC 4-9 lineage? It would be much nicer to scaffold the information of NRC evolution in a broad context.

2) It would be really helpful if the authors could further articulate the reasons of picking the NRC3 clade to further elaborate network topology, which seems to be coming from the result in Fig. S2. What is the relative position of NRC3-driven network in the whole NRC immune network? Is it a part of the network particularly prone to show peripheral modification, if the authors talk about architecture?

3) When the NRC helper network is first introduced in L58-60, it would be advisable to indicate where as it operates, such as within Solanaceae, instead of indicating it later (e.g. L66).

4) Figure 3 itself and the legend alone cannot indicate which color of the bar indicate which protein. Fig. S8A should be moved to the main to inform the readers.

5) The location of six residues (Fig. S17A) shall be introduced in the main figure. From L308 on when the position in number begins to be introduced, till the accompanying text of Fig. 4, the explanation relies on the Sup S17A .

6) L71-73 semi-colon usage seems more adequate grammatically, as current clauses lack a conjunction in some places and the meaning is not clear. For example, was the work of Prf and Rx tested also in N. benthamiana?

7) Fig S7 and accompanying test in the main: please clarify what the curation means. It would be good to see an alignment with SINRC3 to compare this distant copy to the focal one.

8) Fig. S5: ancestral reconstruction of these NRC sequences would heavily depend on the quality of the phylogenetic trees. At least, aligning them with the focal ones, SI and Nb NRCs would point out a distinct or highlight the places with high divergence.

Minor textual comments

L141: “functional divergence has evolved” shall be revised. It is the NRC3 homologs that have evolved to become functionally divergent.

L176: rephrasing is recommended, such that “which molecular events contributed to functional divergence of NRC3”.

L192: this group of NRC3 has “been” nonfunctionalized…

L189-193: this result should be explained with indication of N88 DV used as a positive control.

L197-203: without seeing Fig. S24, it was impossible to obtain information about Petunia NRC3. Please indicate where it belongs.

L284-5: Not all NRCs in Fig.1C is “tested above” experimentally.

L370-372: isn’t it that the Rpi-blb2 decoration with Streptavidin was detected after IPing? The text shall align with the experimental procedure.

The epitope tag flag should be capitalized throughout the manuscript.

**Have all data underlying the figures and results presented in the manuscript been provided?**

Reviewer #1: Yes

Reviewer #2: Yes

Reviewer #3: Yes

PLOS authors have the option to publish the peer review history of their article (what does this mean?). If published, this will include your full peer review and any attached files.

Reviewer #1: No

Reviewer #2: No

Reviewer #3: **Yes: **Eunyoung Chae

---

## [Decision Letter · Decision Letter 1]

26 Aug 2024

Dear Dr Wu,

We are pleased to inform you that your manuscript entitled "Subfunctionalization of NRC3 altered the genetic structure of the Nicotiana NRC network" has been editorially accepted for publication in PLOS Genetics. Congratulations!

Yours sincerely,

Tiancong Qi

Academic Editor

PLOS Genetics

Tanja Slotte

Section Editor

PLOS Genetics

Comments from the reviewers (if applicable):

Reviewer's Responses to Questions

**Comments to the Authors:**

Reviewer #1: The authors have now substantially worked on the text and figures of the manuscript, and have considered all the comments/suggestions made by the reviewers. Especially the concern regarding their interpretation of a transient interaction of the sensor NLR with the helper NRC3. I appreciate this a lot and think that this does not affect the impact and value of the manuscript in a negative way. There is still a huge amount of work in the story and the results are of interest to the field of plant NLR biologists. The discussion is short but sharp and provides the reader a clear hypothesis of how sensor NLRs might activate the NRC helpers - here specifically the NRC3.

This reviewer has no further issues and is looking forward to see this work published.

Reviewer #2: Thank you for your efforts in addressing the concerns I raised in my peer review comments. I appreciate the thorough revisions and the additional clarity you have provided in the revised manuscript. Overall, I am satisfied with your revised version.

Reviewer #3: The authors made a substantial restructuring of the manuscript, which reads very well and highlights main findings. All my comments were adequately addressed point-by-point. Look forward to reading it online.

**Have all data underlying the figures and results presented in the manuscript been provided?**

Reviewer #1: Yes

Reviewer #2: Yes

Reviewer #3: Yes

PLOS authors have the option to publish the peer review history of their article (what does this mean?). If published, this will include your full peer review and any attached files.

Reviewer #1: No

Reviewer #2: No

Reviewer #3: No

**Data Deposition**

http://datadryad.org/submit?journalID=pgenetics&manu=PGENETICS-D-24-00215R1

**Press Queries**

---

## [Editor Report · Acceptance letter]

6 Sep 2024

PGENETICS-D-24-00215R1 

Subfunctionalization of NRC3 altered the genetic structure of the Nicotiana NRC network 

Dear Dr Wu, 

We are pleased to inform you that your manuscript entitled "Subfunctionalization of NRC3 altered the genetic structure of the Nicotiana NRC network" has been formally accepted for publication in PLOS Genetics! Your manuscript is now with our production department and you will be notified of the publication date in due course.

With kind regards,

Anita Estes

PLOS Genetics

On behalf of:
